# Data-Efficient Structured Pruning via Submodular Optimization

**Marwa El Halabi***
Samsung - SAIT AI Lab, Montreal

**Suraj Srinivas†**
Harvard University

**Simon Lacoste-Julien**
Mila, Université de Montreal
Samsung - SAIT AI Lab, Montreal
Canada CIFAR AI Chair

## Abstract

Structured pruning is an effective approach for compressing large pre-trained neural networks without significantly affecting their performance. However, most current structured pruning methods do not provide any performance guarantees, and often require fine-tuning, which makes them inapplicable in the limited-data regime. We propose a principled data-efficient structured pruning method based on submodular optimization. In particular, for a given layer, we select neurons/channels to prune and corresponding new weights for the next layer, that minimize the change in the next layer's input induced by pruning. We show that this selection problem is a weakly submodular maximization problem, thus it can be provably approximated using an efficient greedy algorithm. Our method is guaranteed to have an exponentially decreasing error between the original model and the pruned model outputs w.r.t the pruned size, under reasonable assumptions. It is also one of the few methods in the literature that uses only a limited-number of training data and no labels. Our experimental results demonstrate that our method outperforms state-of-the-art methods in the limited-data regime.

## 1 Introduction

As modern neural networks (NN) grow increasingly large, with some models reaching billions of parameters [McGuffie and Newhouse, 2020], they require an increasingly large amount of memory, power, hardware, and inference time, which makes it necessary to compress them. This is especially important for models deployed on resource-constrained devices like mobile phones and smart speakers, and for latency-critical applications such as self-driving cars.

Several approaches exist to compress NNs. Some methods approximate model weights using quantization and hashing [Gong et al., 2014, Courbariaux et al., 2015], or low-rank approximation and tensor factorization [Denil et al., 2013, Lebedev et al., 2015, Su et al., 2018]. In another class of methods called knowledge distillation, a small network is trained to mimic a much larger network [Bucila et al., 2006, Hinton et al., 2015]. Other methods employ sparsity and group-sparsity regularisation during training, to induce sparse weights [Collins and Kohli, 2014, Voita et al., 2019].

In this work, we follow the network pruning approach, where the redundant units (weights, neurons or filters/channels) of a pre-trained NN are removed; see [Kuzmin et al., 2019, Blalock et al., 2020,

---

*work done partially at MIT, CSAIL.
†work done partially at Idiap Research Institute, Switzerland.

36th Conference on Neural Information Processing Systems (NeurIPS 2022).

[Hoefler et al., 2021] for recent surveys. We also focus on the limited-data regime, where only few training data is available and data labels are unavailable. The advantage of pruning approaches is that, unlike weights approximation-based methods, they preserve the network structure, allowing retraining after compression, and unlike training-based approaches, they do not require training from scratch, which is costly and requires large training data. It is also possible to combine different compression approaches to compound their benefits, see e.g., [Kuzmin et al., 2019, Section 4.3.4].

Existing pruning methods fall into two main categories: unstructured pruning methods which prune individual weights leading to irregular sparsity patterns, and structured pruning methods which prune regular regions of weights, such as neurons, channels, or attention heads. Structured pruning methods are generally preferable as the resulting pruned models can work with off-the-shelf hardware or kernels, as opposed to models pruned with unstructured pruning which require specialized ones.

The majority of existing structured pruning methods are heuristics that do not offer any theoretical guarantees. Moreover, most pruning methods are inapplicable in the limited-data regime, as they rely on fine-tuning with large training data for at least a few epochs to recover some of the accuracy lost with pruning. [Mariet and Sra, 2015] proposed a "reweighting" procedure applicable to any pruning method, which optimize the remaining weights of the next layer to minimize the change in the input to the next layer. Their empirical results on pruning single linear layers suggest that reweighting can provide a similar boost to performance as fine-tuning, without the need for data labels.

**Our contributions**  We propose a principled data-efficient structured pruning method based on submodular optimization. In each layer, our method simultaneously selects neurons to prune and new weights for the next layer, that minimize the change in the next layer's input induced by pruning. The optimization with respect to the weights, for a *fixed* selection of neurons, is the same one used for reweighting in [Mariet and Sra, 2015]. The resulting subset selection problem is intractable, but we show that it can be formulated as a weakly submodular maximization problem (see Definition 2.1). We can thus use the standard greedy algorithm to obtain a $(1 - e^{-\gamma})$-approximation to the optimal solution, where $\gamma$ is non-zero if we use sufficient training data. We further adapt our method to prune any regular regions of weights; we focus in particular on pruning channels in convolution layers. To prune multiple layers in the network, we apply our method to each layer independently or sequentially.

We show that the error induced by pruning with our method on the model output decays with an $O(e^{-\gamma k})$ rate w.r.t the number $k$ of neurons/channels kept, under reasonable assumptions. Our method uses only limited training data and no labels. Similar to [Mariet and Sra, 2015], we observe that reweighting provides a significant boost in performance not only to our method, but also to other baselines we consider. However unlike [Mariet and Sra, 2015], we only use a small fraction of the training data, around $\sim 1\%$ in our experiments. Our experimental results demonstrate that our method outperforms state-of-the-art pruning methods, even when reweighting is applied to them too, in the limited-data regime, and it is among the best performing methods in the standard setting.

**Related work**  A large variety of structured pruning approaches has been proposed in the literature, based on different selection schemes and algorithms to solve them. Some works prune neurons/channels individually based on some importance score [He et al., 2014, Li et al., 2017, Liebenwein et al., 2020, Mussay et al., 2020, 2021, Molchanov et al., 2017, Srinivas and Babu, 2015]. Such methods are efficient and easy to implement, but they fail to capture higher-order interactions between the pruned parameters. Most do not provide any performance guarantee. One exception are the sampling-based methods of [Liebenwein et al., 2020, Mussay et al., 2020, 2021], who show an $O(1/k)$ error rate, under some assumptions on the model activations.

Closer to our approach are methods that aim to prune neurons/channels that minimize the change induced by pruning in the output of the layer being pruned, or its input to the next layer [Luo et al., 2017, He et al., 2017, Zhuang et al., 2018, Ye et al., 2020b]. These criteria yield an intractable combinatorial problem. Existing methods either use a heuristic greedy algorithm to solve it [Luo et al., 2017, Zhuang et al., 2018], or they solve instead its $\ell_1$-relaxation using alternating minimization [He et al., 2017], or a greedy algorithm with Frank-Wolfe like updates [Ye et al., 2020b]. Among these works only [Ye et al., 2020b] provides theoretical guarantees, showing an $O(e^{-ck})$ error rate. Their method is more expensive than ours, and only optimize the scaling of the next layer weights instead of the weights themselves. A global variant of this method is proposed in Ye et al. [2020a,b], which aim to prune neurons/channels that directly minimize the loss of the pruned network. A similar greedy algorithm with Frank-Wolfe like updates is used to solve the $\ell_1$-relaxation of the selection problem. This method has an $O(1/k^2)$ error rate and is very expensive, as it requires a full forward

pass through the network at each iteration. See Appendix A for a more detailed comparison of our method with those of [Ye et al., 2020a,b].

Mariet and Sra [2015] depart from the usual strategy of pruning parameters whose removal influences the network the least. They instead select a subset of diverse neurons to keep in each layer by sampling from a Determinantal Point Process, then they apply their reweighting procedure. Their experimental results show that the advantage of their method is mostly due to reweighting (see Figure 4 therein).

## 2 Preliminaries

We begin by introducing our notation and some relevant background from submodular optimization.

**Notation:** Given a ground set $V = \{1, 2, \cdots, d\}$ and a set function $F : 2^V \to \mathbb{R}_+$, we denote the *marginal gain* of adding a set $I \subseteq V$ to another set $S \subseteq V$ by $F(I \mid S) = F(S \cup I) - F(S)$, which quantifies the change in value when adding $I$ to $S$. The cardinality of a set $S$ is written as $|S|$. Given a vector $x \in \mathbb{R}^d$, we denote its support set by $\text{supp}(x) = \{i \in V | x_i \neq 0\}$, and its $\ell_2$-norm by $\|x\|_2$. Given a matrix $X \in \mathbb{R}^{d' \times d}$, we denote its $i$-th column by $X_i$, and its Frobenius norm by $\|X\|_F$. Given a set $S \subseteq V$, $X_S$ is the matrix with columns $X_i$ for all $i \in S$, and 0 otherwise, and $\mathbf{1}_S$ is the indicator vector of $S$, with $[\mathbf{1}_S]_i = 1$ for all $i \in S$, and 0 otherwise.

---

**Algorithm 1** GREEDY

1: **Input:** Ground set $V$, set function $F : 2^V \to \mathbb{R}_+$, budget $k \in \mathbb{N}_+$
2: $S \leftarrow \emptyset$
3: **while** $|S| < k$ **do**
4:     $i^* \leftarrow \arg\max_{i \in V \setminus S} F(i \mid S)$
5:     $S \leftarrow S \cup \{i^*\}$
6: **end while**
7: **Output:** $S$

---

**Weakly submodular maximization:** A set function $F$ is *submodular* if it has diminishing marginal gains: $F(i \mid S) \geq F(i \mid T)$ for all $S \subseteq T$, $i \in V \setminus T$. We say that $F$ is *normalized* if $F(\emptyset) = 0$, and non-decreasing if $F(S) \leq F(T)$ for all $S \subseteq T$.
Given a non-decreasing submodular function $F$, selecting a set $S \subseteq V$ with cardinality $|S| \leq k$ that maximize $F(S)$ can be done efficiently using the GREEDY algorithm (Alg. 1). The returned solution is guaranteed to satisfy $F(\hat{S}) \geq (1 - 1/e) \max_{|S| \leq k} F(S)$ [Nemhauser et al., 1978]. In general though maximizing a non-submodular function over a cardinality constraint is NP-Hard [Natarajan, 1995]. However, Das and Kempe [2011] introduced a notion of *weak submodularity* which is sufficient to obtain a constant factor approximation with the GREEDY algorithm.

**Definition 2.1.** Given a set function $F : 2^V \to \mathbb{R}$, $U \subseteq V$, $k \in \mathbb{N}_+$, we say that $F$ is $\gamma_{U,k}$-weakly submodular, with $\gamma_{U,k} > 0$ if

$$\gamma_{U,k} F(S|L) \leq \sum_{i \in S} F(i|L),$$

for every two disjoint sets $L, S \subseteq V$, such that $L \subseteq U, |S| \leq k$.

The parameter $\gamma_{U,k}$ is called the *submodularity ratio* of $F$. It characterizes how close a set function is to being submodular. If $F$ is non-decreasing then $\gamma_{U,k} \in [0, 1]$, and $F$ is submodular if and only if $\gamma_{U,k} = 1$ for all $U \subseteq V, k \in \mathbb{N}_+$. Given a non-decreasing $\gamma_{\hat{S},k}$-weakly submodular function $F$, the Greedy algorithm is guaranteed to return a solution $\hat{S}$ satisfying $F(\hat{S}) \geq (1 - e^{-\gamma_{\hat{S},k}}) \max_{|S| \leq k} F(S)$ [Elenberg et al., 2016, Das and Kempe, 2011]. Hence, the closer $F$ is to being submodular, the better is the approximation guarantee.

## 3 Reweighted input change pruning

In this section, we introduce our approach for pruning neurons in a single layer. Given a large pretrained NN, $n$ training data samples, and a layer $\ell$ with $n_\ell$ neurons, our goal is to select a small number

$k$ out of the $n_\ell$ neurons to keep, and prune the rest, in a way that influences the network the least. One way to achieve this is by minimizing the change in input to the next layer $\ell + 1$, induced by pruning. However, simply throwing away the activations from the dropped neurons is wasteful. Instead, we optimize the weights of the next layer to reconstruct the inputs from the remaining neurons.

Formally, let $A^\ell \in \mathbb{R}^{n \times n_\ell}$ be the activation matrix of layer $\ell$ with columns $a_1^\ell, \cdots, a_{n_\ell}^\ell$, where $a_i^\ell \in \mathbb{R}^n$ is the vector of activations of the $i$th neuron in layer $\ell$ for each training input, and let $W^{\ell+1} \in \mathbb{R}^{n_\ell \times n_{\ell+1}}$ be the weight matrix of layer $\ell + 1$ with columns $w_1^{\ell+1}, \cdots, w_{n_{\ell+1}}^{\ell+1}$, where $w_i^{\ell+1} \in \mathbb{R}^{n_\ell}$ is the vector of weights connecting the $i$th neuron in layer $\ell + 1$ to the neurons in layer $\ell$. When a neuron is pruned in layer $\ell$, the corresponding column of weights in $W^\ell$ and row in $W^{\ell+1}$ are removed. Pruning $n_\ell - k$ neurons in layer $\ell$ reduces the number of parameters and computation cost by $(n_\ell - k)/n_\ell$ for both layer $\ell$ and $\ell + 1$.

Let $V_\ell = \{1, \cdots, n_\ell\}$. Given a set $S \subseteq V_\ell$, we denote by $A_S^\ell$ the matrix with columns $a_i^\ell$ for all $i \in S$, and 0 otherwise. That is, $A_S^\ell$ is the activation matrix of layer $\ell$ after pruning. We choose a set of neurons $S \subseteq V_\ell$ to keep and new weights $\tilde{W}^{\ell+1} \in \mathbb{R}^{n_\ell \times n_{\ell+1}}$ that minimize:

$$\min_{|S| \leq k, \tilde{W}^{\ell+1} \in \mathbb{R}^{n_\ell \times n_{\ell+1}}} \|A^\ell W^{\ell+1} - A_S^\ell \tilde{W}^{\ell+1}\|_F^2 \tag{1}$$

Note that $A^\ell W^{\ell+1}$ are the original inputs of layer $l + 1$, and $A_S^\ell \tilde{W}^{\ell+1}$ are the inputs after pruning and reweighting, i.e., replacing the weights $W^{\ell+1}$ of layer $\ell + 1$ with the new weights $\tilde{W}^{\ell+1}$.

## 3.1 Greedy selection

Solving Problem (1) exactly is NP-Hard [Natarajan, 1995]. However, we show below that it can be formulated as a weakly submodular maximization problem, hence it can be efficiently approximated. Let

$$F(S) = \|A^\ell W^{\ell+1}\|_F^2 - \min_{\tilde{W}^{\ell+1}} \|A^\ell W^{\ell+1} - A_S^\ell \tilde{W}^{\ell+1}\|_F^2, \tag{2}$$

then Problem (1) is equivalent to $\max_{|S| \leq k} F(S)$.

**Proposition 3.1.** *Given $U \subseteq V, k \in \mathbb{N}_+$, $F$ is a normalized non-decreasing $\gamma_{U,k}$-weakly submodular function, with*

$$\gamma_{U,k} \geq \frac{\min_{\|z\|_2=1, \|z\|_0 \leq |U|+k} \|A^\ell z\|_2^2}{\max_{\|z\|_2=1, \|z\|_0 \leq |U|+1} \|A^\ell z\|_2^2}.$$

The proof of Proposition 3.1 follows by writing $F$ as the sum of $n_{\ell+1}$ sparse linear regression problems $F(S) = \sum_{m=1}^{n_{\ell+1}} \|A^\ell w_m^{\ell+1}\|_2^2 - \min_{\text{supp}(\tilde{w}_m) \subseteq S} \|A^\ell w_m^{\ell+1} - A^\ell \tilde{w}_m\|_2^2$, and from the relation established in [Elenberg et al., 2016, Das and Kempe, 2011] between weak submodularity and sparse eigenvalues of the covariance matrix (see Appendix B.1).

We use the GREEDY algorithm to select a set $\hat{S} \subseteq V_\ell$ of $k$ neurons to keep in layer $\ell$. As discussed in Section 2, the returned solution is guaranteed to satisfy

$$F(\hat{S}) \geq (1 - e^{-\gamma_{\hat{S},k}}) \max_{|S| \leq k} F(S) \tag{3}$$

Computing the lower bound on the submodularity ratio $\gamma_{\hat{S},k}$ in Proposition 3.1 is NP-Hard [Das and Kempe, 2011]. It is non-zero if any $\min\{2k, n_\ell\}$ columns of $A^\ell$ are linearly independent. If the number of training data is larger than the number of neurons, i.e., $n > n_\ell$, this is likely to be satisfied. We verify that this is indeed the case in our experiments in Appendix E. We also discuss the tightness of the lower bound in Appendix F.

We show in Appendix D that $F$ satisfies an even stronger notion of approximate submodularity than weak submodularity, which implies a better approximation guarantee for GREEDY than the one provided in Eq. (3). Though, this requires a stronger assumption: any $k + 1$ columns of $A^\ell$ should be linearly independent and all rows of $W^{\ell+1}$ should be linearly independent. In particular, we would need that $n_\ell \leq n_{\ell+1}$, which is not always satisfied.

In Section 6, we show that the approximation guarantee of Greedy implies an exponentially decreasing bound on the layerwise error, and on the final output error under a mild assumption.

## 3.2 Reweighting

For a fixed $S \subseteq V_\ell$, the reweighted input change $\|A^\ell W^{\ell+1} - A_S^\ell \tilde{W}^{\ell+1}\|_F^2$ is minimized by setting

$$\tilde{W}^{\ell+1} = x^S(A^\ell)W^{\ell+1}, \tag{4}$$

where $x^S(A^\ell) \in \mathbb{R}^{n_\ell \times n_\ell}$ is the matrix with columns $x^S(a_j^\ell)$ such that

$$x^S(a_j^\ell) \in \operatorname*{arg\,min}_{\operatorname{supp}(x) \subseteq S} \|a_j^\ell - Ax\|_2^2 \text{ for all } j \in V_\ell. \tag{5}$$

Note that the new weights are given by $\tilde{w}_{im}^{\ell+1} = w_{im}^{\ell+1} + \sum_{j \notin S}[x^S(A^\ell)]_{ij}w_{jm}^{\ell+1}$ for all $i \in S$, and $\tilde{w}_{im}^{\ell+1} = 0$ for all $i \notin S, m \in V_{\ell+1}$. Namely, the new weights merge the weights from the dropped neurons into the kept ones. This is the same reweighting procedure introduced in [Mariet and Sra, 2015]. But instead of applying it only at the end to the selected neurons $\hat{S}$, it is implicitly done at each iteration of our pruning method, as it is required to evaluate $F$. We discuss next how this can be done efficiently.

## 3.3 Cost

Each iteration of GREEDY requires $O(n_\ell)$ function evaluations of $F$. Computing $F(S)$ from scratch needs $O(k \cdot (n_\ell \cdot n_{\ell+1} + n \cdot (n_\ell + n_{\ell+1})))$ time, so a naive implementation of GREEDY is too expensive. The following Proposition outlines how we can efficiently evaluate $F(S+i)$ given that $F(S)$ was computed in the previous iteration.

**Proposition 3.2.** *Given $S \subseteq V_\ell$ such that $|S| \leq k$, $i \notin S$, let $\operatorname{proj}_S(a_j^\ell) = A_S^\ell x^S(a_j^\ell)$ be the projection of $a_j^\ell$ onto the column space of $A_S^\ell$, $R_S(a_i^\ell) = a_i^\ell - \operatorname{proj}_S(a_i^\ell)$ and $\operatorname{proj}_{R_S(a_i^\ell)}(a_j^\ell) \in \arg\min_{z=R_S(a_i^\ell)\gamma,\gamma\in\mathbb{R}} \|a_j^\ell - z\|_2^2$ the corresponding residual and the projection of $a_j^\ell$ onto it. We can write*

$$F(i|S) = \sum_{m=1}^{n_{\ell+1}} \|\operatorname{proj}_{R_S(a_i^\ell)}(A_{V\setminus S}^\ell)w_m^{\ell+1}\|_2^2,$$

*where $\operatorname{proj}_{R_S(a_i)}(A_{V\setminus S}^\ell)$ is the matrix with columns $\operatorname{proj}_{R_S(a_i)}(a_j^\ell)$ for all $j \notin S$, $0$ otherwise. Assuming $F(S), \operatorname{proj}_S(a_j^\ell)$ and $x^S(a_j^\ell)$ for all $j \notin S$ were computed in the previous iteration, we can compute $F(S+i), \operatorname{proj}_{S+i}(a_j^\ell)$ and $x^{S+i}(a_j^\ell)$ for all $j \notin (S+i)$ in*

$$O(n_\ell \cdot (n_{\ell+1} + n + k)) \text{ time.}$$

*The optimal weights in Eq. (4) can then be computed in $O(k \cdot n_\ell \cdot n_{\ell+1})$ time, at the end of GREEDY.*

The proof is given in Appendix B.2, and relies on using optimality conditions to construct the least squares solution $x^{S+i}(a_j^\ell)$ from $x^S(a_j^\ell)$.

In total GREEDY's runtime is then $O(k \cdot (n_\ell)^2 \cdot (n_{\ell+1} + n + k))$. In other words, our pruning method costs as much as $O(k)$ forward passes in layer $\ell+1$ with a batch of size $n$ (assuming $n_{\ell+1} = O(n_\ell)$). Using a faster variant of GREEDY, called STOCHASTIC-GREEDY [Mirzasoleiman et al., 2015], further reduces the cost to $O(\log(1/\epsilon) \cdot (n_\ell)^2 \cdot (n_{\ell+1} + n + k))$, or equivalently $O(\log(1/\epsilon))$ forward passes in layer $\ell+1$ with a batch of size $n$, while maintaining almost the same approximation guarantee $(1 - e^{-\gamma_{\hat{S},k}} - \epsilon)$ in expectation. [3]

Note also that computing the solutions for different budgets $k' \leq k$ can be done at the cost of one by running GREEDY with budget $k$. Our method is more expensive than methods which prune neurons individually [He et al., 2014, Li et al., 2017, Liebenwein et al., 2020, Mussay et al., 2020, 2021, Molchanov et al., 2017, Srinivas and Babu, 2015], but much less expensive than a loss-based method like [Ye et al., 2020a,b], which requires $O(k)$ forward passes in the full network, for each layer.

## 4 Pruning regular regions of neurons

In this section, we discuss how to adapt our approach to pruning regular regions of neurons. This is easily achieved by mapping any set of regular regions to the corresponding set of neurons, then applying the same method in Section 3. In particular, we focus on pruning channels in CNNs.

---

[3]Mirzasoleiman et al. [2015] only consider submodular functions, but it is straighforward to extend their result to weakly submodular functions Appendix B.3.

Given a layer $\ell$ with $n_\ell$ output channels, let $X^\ell \in \mathbb{R}^{n \cdot p_\ell \times n_\ell \times r_h \times r_w}$ be its activations for each output channel and training input, where $p_\ell$ is number of patches obtained by applying a filter of size $r_h \times r_w$, and let $F^{\ell+1} \in \mathbb{R}^{n_{\ell+1} \times n_\ell \times r_h \times r_w}$ be the weights of layer $\ell + 1$, corresponding to $n_\ell$ filters of size $r_h \times r_w$ for each of its output channels. When an output channel is pruned in layer $\ell$, the corresponding weights in $F^\ell$ and $F^{\ell+1}$ are removed. Pruning $n_\ell - k$ output channels in layer $\ell$ reduces the number of parameters and computation cost by $(n_\ell - k)/n_\ell$ for both layer $\ell$ and $\ell + 1$. If layer $\ell$ is followed by a batch norm layer, the weights therein corresponding to the pruned channels are also removed.

We arrange the activations $X_c^\ell \in \mathbb{R}^{n \cdot p_\ell \times r_h \cdot r_w}$ of each channel $c$ into $r_h r_w$ columns of $A^\ell \in \mathbb{R}^{n \cdot p_\ell \times n_\ell \cdot r_h \cdot r_w}$, i.e., $A^\ell = [X_1^\ell, \cdots, X_{n_\ell}^\ell]$. Similarly, we arrange the weights $F_c^{\ell+1} \in \mathbb{R}^{n_{\ell+1} \times r_h \times r_w}$ of each channel $c$ into $r_h \cdot r_w$ rows of $W^{\ell+1} \in \mathbb{R}^{n_\ell \cdot r_h \cdot r_w \times n_{\ell+1}}$, i.e., $(W^{\ell+1})^\top = [(F_1^\ell)^\top, \cdots, (F_{n_\ell}^\ell)^\top]$. Recall that $V_\ell = \{1, \cdots, n_\ell\}$, and let $V_\ell' = \{1, \cdots, r_h r_w n_\ell\}$. We define a function $M : 2^{V_\ell} \to 2^{V_\ell'}$ which maps every channel $c$ to its corresponding $r_h r_w$ columns in $A^\ell$. Let $G(S) = F(M(S))$, with $F$ defined in Eq. (2), then minimizing the reweighted input change $\|A^\ell W^{\ell+1} - A_{M(S)}^\ell \tilde{W}^{\ell+1}\|_F^2$ with a budget $k$ is equivalent to $\max_{|S| \le k} G(S)$. The following proposition shows that this remains a weakly submodular maximization problem.

**Proposition 4.1.** *Given $U \subseteq V_\ell, k \in \mathbb{N}_+$, $G$ is a normalized non-decreasing $\gamma_{U,k}$-weakly submodular function, with*

$$\gamma_{U,k} \ge \frac{\min_{\|z\|_2=1, \|z\|_0 \le r_h r_w(|U|+k)} \|A^\ell z\|_2^2}{\max_{\|z\|_2=1, \|z\|_0 \le r_h r_w(|U|+1)} \|A^\ell z\|_2^2}.$$

*Proof sketch.* $G$ is $\gamma_{U,k}$-weakly submodular iff $F$ satisfies $\gamma_{U,k} F(M(S)|M(L)) \le \sum_{i \in S} F(M(i)|M(L))$, for every two disjoint sets $L, S \subseteq V_\ell$, such that $L \subseteq U, |S| \le k$. The proof follows by extending the relation established in [Elenberg et al., 2016, Das and Kempe, 2011] between weak submodularity and sparse eigenvalues of the covariance matrix to this case. $\square$

As before, we use the GREEDY algorithm, with function $G$, to select a set $\hat{S} \subseteq V_\ell$ of $k$ channels to keep in layer $\ell$. We get the same approximation guarantee $G(\hat{S}) \ge (1 - e^{-\gamma_{\hat{S},k}}) \max_{|S| \le k} G(S)$. The submodularity ratio $\gamma_{\hat{S},k}$ is non-zero if any $\min\{2k, n_\ell\} r_h r_w$ columns of $A^\ell$ are linearly independent. In our experiments, we observe that in certain layers linear independence only holds for $k$ very small, e.g., $k \le 0.01 n_\ell$. This is due to the correlation between patches which overlap. To remedy this, we experimented with using only $r_h r_w$ random patches from each image, instead of using all patches. This indeed raises the rank of $A^\ell$, but certain layers have a very small feature map size so that even the small number of random patches have significant overlap, resulting in still a very small range where linear independence holds, e.g., $k \le 0.08 n_\ell$ (see Appendix E for more details). The results obtained with random patches were worst than the ones with all patches, we thus omit them. Note that our lower bounds on $\gamma_{\hat{S},k}$ are not necessarily tight (see Appendix F). Hence, having linear dependence does not necessarily imply that $\gamma_{\hat{S},k} = 0$; our method still performs well in these cases.

For a fixed $S \subseteq V_\ell$, the optimal weights are again given by $\tilde{W}^{\ell+1} = x^{M(S)}(A^\ell) W^{\ell+1}$. The cost of running GREEDY and reweighting is the same as before (see Appendix B.2).

## 5 Pruning multiple layers

In this section, we explain how to apply our pruning method to prune multiple layers of a NN.

### 5.1 Reweighted input change pruning variants

We consider three variants of our method: LAYERINCHANGE, SEQINCHANGE, and ASYMIN-CHANGE. In LAYERINCHANGE, we prune each layer independently, i.e., we apply exactly the method in Section 3 or 4, according to the layer's type. This is the fastest variant; it has the same cost as pruning a single layer, as each layer can be pruned in parallel, and it only requires one forward pass to get the activations of all layers. However, it does not take into account the effect of pruning one layer on subsequent layers.

In SEQINCHANGE, we prune each layer sequentially, starting from the earliest layer to the latest one. For each layer $\ell$, we apply our method with $A^\ell$ replaced by the *updated* activations $B^\ell$ after

having pruned previous layers, i.e., we solve $\min_{|S|\leq k, \tilde{W}^{\ell+1}\in\mathbb{R}^{n_\ell\times n_{\ell+1}}} \|B^\ell W^{\ell+1} - B_S^\ell \tilde{W}^{\ell+1}\|_F^2$. In ASYMINCHANGE, we also prune each layer sequentially, but to avoid the accumulation of error, we use an asymmetric formulation of the reweighted input change, where instead of approximating the *updated* input $B^\ell W^{\ell+1}$, we approximate the *original* input $A^\ell W^{\ell+1}$, i.e., we solve $\min_{|S|\leq k, \tilde{W}^{\ell+1}\in\mathbb{R}^{n_\ell\times n_{\ell+1}}} \|A^\ell W^{\ell+1} - B_S^\ell \tilde{W}^{\ell+1}\|_F^2$. This problem is still a weakly submodular maximization problem, with the same submodularity ratio given in Propositions 3.1 and 4.1, with $A^\ell$ replaced by $B^\ell$ therein (see Appendix B.1). Hence, the same approximation guarantee as in the symmetric formulation holds here. Moreover, a better approximation guarantee can again be obtained under stronger assumptions (see Appendix D). The cost of running GREEDY with the asymmetric formulation and reweighting is also the same as before (see Appendix B.2).

In Section 6, we show that the sequential variants of our method both have an exponential error rate, which is faster for the asymmetric variant. We evaluate all three variants in our experiments. As expected, ASYMINCHANGE usually performs the best, and LAYERINCHANGE the worst.

## 5.2 Per-layer budget selection

Another important design choice is how much to prune in each layer, given a desired global compression ratio (see Appendix I for the effect of this choice on performance). In our experiments, we use the budget selection method introduced in [Kuzmin et al., 2019, Section 3.4.1], which can be applied to any layerwise pruning method, thus enabling us to have a fair comparison.

Given a network with $L$ layers to prune, let $c = \frac{\text{original size}}{\text{pruned size}}$ be the desired compression ratio. We want to select for each layer $\ell$, the number of neurons/channels $k_\ell = \alpha_\ell n_\ell$ to keep, with $\alpha_\ell$ chosen from a fixed set of possible values, e.g., $\alpha_\ell \in \{0.05, 0.1, \cdots, 1\}$. We define a layerwise accuracy metric $P_\ell(k_\ell)$ as the accuracy obtained after pruning layer $\ell$, with a budget $k_\ell$, while other layers are kept intact, evaluated on a verification set. We set aside a subset of the training set to use as a verification set. Let $P_{\text{orig}}$ be the original model accuracy, $C_{\text{orig}}$ the original model size, and $C(k_1, \cdots, k_L)$ the pruned model size. We select the per-layer budgets that minimize the per-layer accuracy drop while satisfying the required compression ratio:

$$\min_{k_1,\cdots,k_L} \{\tau : \forall \ell \in [L], P_\ell(k_\ell) \geq P_{\text{orig}} - \tau, C(k_1, \cdots, k_L) \leq C_{\text{orig}}/c\}. \tag{6}$$

We can solve the selection problem (6) using binary search, if the layerwise accuracy $P_\ell(k_\ell)$ is a non-decreasing function of $k_\ell$. Empirically, this is not always the case, the general trend is non-decreasing, but some fluctuations occur. In such cases, we use interpolation to ensure monotonicity.

Alternatively, another simple strategy is to prune each layer until the perlayer error (the reweighted input change in our case) reaches some threshold $\epsilon$, and vary $\epsilon$ to obtain the desired compression ratio, as done in [Zhuang et al., 2018, Ye et al., 2020a].

# 6 Error convergence rate

In this section, we provide the error rate of our proposed method. The omitted proofs are given in Appendix C. We first show that the change in input to the next layer induced by pruning with our method, with both the symmetric and asymmetric formulation, decays with exponentially fast rate.

**Proposition 6.1.** *Let $\hat{S}$ be the output of the GREEDY algorithm and $\hat{W}^{\ell+1}$ the corresponding optimal weights (Eq. (4)), then*

$$\|A^\ell W^{\ell+1} - A_{\hat{S}}^\ell \hat{W}^{\ell+1}\|_F^2 \leq e^{-\gamma_{\hat{S},n_\ell} k/n_\ell} \|A^\ell W^{\ell+1}\|_F^2,$$

*and*

$$\|A^\ell W^{\ell+1} - B_{\hat{S}}^\ell \hat{W}^{\ell+1}\|_F^2 \leq e^{-\gamma_{\hat{S},n_\ell} k/n_\ell} \|A^\ell W^{\ell+1}\|_F^2 + (1 - e^{-\gamma_{\hat{S},n_\ell} k/n_\ell}) \min_{\tilde{W}^{\ell+1}\in\mathbb{R}^{n_\ell\times n_{\ell+1}}} \|A^\ell W^{\ell+1} - B^\ell \tilde{W}^{\ell+1}\|_F^2$$

This follows by extending the approximation guarantee of GREEDY in [Elenberg et al., 2016, Das and Kempe, 2011] to $F(\hat{S}) \geq (1 - e^{-\gamma_{\hat{S},n_\ell} k/n_\ell}) \max_{|S|\leq n_\ell} F(S)$. Note that this bounds uses the submodularity ratio $\gamma_{\hat{S},n_\ell}$, for which the lower bound in Proposition 3.1 is non-zero only if *all* columns of $A^\ell$ are linearly independent, which is more restrictive. Though as discussed earlier, this

bound is not necessarily tight. We can further extend this exponential layerwise error rate to an exponentially rate on the final output error, if we assume as in [Ye et al., 2020b] that the function corresponding to all layers coming after layer $\ell$ is Lipschitz continuous.

**Corollary 6.2.** *Let $y \in \mathbb{R}^n$ be the original model output, $y^{\hat{S}} \in \mathbb{R}^n$ the output after layer $\ell$ is pruned using our method, and $H$ the function corresponding to all layers coming after layer $\ell$, i.e., $y = H(A^\ell W^{\ell+1}), y^{\hat{S}} = H(A_{\hat{S}}^\ell \hat{W}^{\ell+1})$. If $H$ is Lipschitz continuous with constant $\|H\|_{Lip}$, then*

$$\|y - y^{\hat{S}}\|_2^2 \leq e^{-\gamma_{\hat{S}, n_\ell} k/n_\ell} \|H\|_{Lip}^2 \|A^\ell W^{\ell+1}\|_F^2.$$

*Proof.* Since $H$ is Lipschitz continuous, we have $\|y - y^{\hat{S}}\|_2^2 \leq \|H\|_{Lip}^2 \|A^\ell W^{\ell+1} - A_{\hat{S}}^\ell \hat{W}^{\ell+1}\|_F^2$. The claim then follows from Proposition 6.1. $\square$

This matches the exponential convergence rate achieved by the local imitation method in [Ye et al., 2020b, Theorem 1], albeit with a different constant. Under the same assumption, we can show that pruning multiple layers with the sequential variants of our method, SEQINCHANGE and ASYMIN-CHANGE, also admits an exponential convergence rate:

**Corollary 6.3.** *Let $y \in \mathbb{R}^n$ be the original model output, $y^{\hat{S}_\ell}, y^{\tilde{S}_\ell} \in \mathbb{R}^n$ the outputs after layers $1$ to $\ell$ are sequentially pruned using SEQINCHANGE and ASYMINCHANGE, respectively, and $H_\ell$ the function corresponding to all (unpruned) layers coming after layer $\ell$. If every function $H_\ell$ is Lipschitz continuous with constant $\|H_\ell\|_{Lip}$, then*

$$\|y - y^{\hat{S}_L}\|_2^2 \leq \sum_{\ell=1}^L e^{-\gamma_{\hat{S}_\ell, n_\ell} k_\ell/n_\ell} \|H_\ell\|_{Lip}^2 \|A^\ell W^{\ell+1}\|_F^2,$$

*and*

$$\|y - y^{\tilde{S}_L}\|_2^2 \leq \sum_{\ell=1}^L \prod_{\ell'=\ell+1}^L (1 - e^{-\gamma_{\tilde{S}_{\ell'}, n_{\ell'}} k_{\ell'}/n_{\ell'}}) e^{-\gamma_{\tilde{S}_\ell, n_\ell} k_\ell/n_\ell} \|H_\ell\|_{Lip}^2 \|A^\ell W^{\ell+1}\|_F^2.$$

The result is obtained by iteratively applying Proposition 6.1 to the error incurred after each layer is pruned. The rate of SEQINCHANGE matches the exponential convergence rate achieved by the local imitation method in [Ye et al., 2020b, Theorem 6]. The bound on ASYMINCHANGE is stronger, confirming that the asymmetric formulation indeed reduces the accumulation of errors.

# 7 Empirical Evaluation

In this section, we examine the performance of our proposed pruning method in the limited-data regime. To that end, we focus on one-shot pruning, in which a pre-trained model is compressed in a single step, without any fine-tuning. We study the effect of fine-tuning with both limited and sufficient data in Appendix H. We compare the three variants of our method, LAYERINCHANGE, SEQINCHANGE, and ASYMINCHANGE, with the following baselines:

- LAYERGREEDYFS [Ye et al., 2020a]: for each layer, first removes all neurons/channels in that layer, then gradually adds back the neuron/channel that yields the largest decrease of the loss, evaluated on one batch of training data. Layers are pruned sequentially from the input to the output layer.

- LAYERSAMPLING [Liebenwein et al., 2020]: samples neurons/channels, in each layer, with probabilities proportional to sensitivities based on (activations $\times$ weights), and prunes the rest.

- ACTGRAD [Molchanov et al., 2017]: prunes neurons/channels with the lowest (activations $\times$ gradients), averaged over the training data, with layerwise $\ell_2$-normalization.

- LAYERACTGRAD: prunes neurons/channels with the lowest (activations $\times$ gradients), averaged over the training data, in each layer. This is the layerwise variant of ACTGRAD.

- LAYERWEIGHTNORM [Li et al., 2017]: prunes neurons/channels with the lowest output weights $\ell_1$-norm, in each layer.

- RANDOM: prunes randomly selected neurons/channels globally across layers in the network.

- LAYERRANDOM: prunes randomly selected neurons/channels in each layer.

We also considered the global variant of LAYERWEIGHTNORM proposed in [He et al., 2014], but we exclude it from plots, as it is always the worst performing method. We evaluate the performance of these methods on the LeNet model [LeCun et al., 1989] on the MNIST dataset [Lecun et al., 1998], and on the ResNet56 [He et al., 2016] and the VGG11 [Simonyan and Zisserman, 2015] models on the CIFAR-10 dataset [Krizhevsky et al., 2009]. To ensure a fair comparison, all experiments are based on our own implementation of all the compared methods. To compute the gradients and activations used for pruning in LAYERSAMPLING, ACTGRAD, LAYERACTGRAD, and our method's variants, we use four batches of 128 training images, i.e., $n = 512$, which corresponds to $\sim 1\%$ of the training data in MNIST and CIFAR10. We consider two variants of the method proposed in [Ye et al., 2020a]: a limited-data variant LAYERGREEDYFS which only uses the same four batches of data used in our method, and a full-data variant LAYERGREEDYFS-fd with access to the full training data.

We report top-1 accuracy results evaluated on the validation set, as we vary the compression ratio ($\frac{\text{original size}}{\text{pruned size}}$). Unless otherwise specified, we use the per-layer budget selection method described in Section 5.2 for all the layerwise pruning methods, except for LAYERSAMPLING for which we use its own budget selection strategy provided in [Liebenwein et al., 2020]. We use a subset of the training set, of the same size as the validation set, as a verification set for the budget selection method. To disentangle the benefit of using our pruning method from the benefit of reweighting (Section 3.2), we report results with reweighting applied to all pruning methods, or none of them. Though, we will focus our analysis on the more interesting results with reweighting, with the plots without reweighting mostly serving as a demonstration of the benefit of reweighting. Results are averaged over five random runs, with standard deviations plotted as error bars. We report the speedup ($\frac{\text{original number of FLOPs}}{\text{pruned number of FLOPs}}$) and pruning time values in Appendix J. For additional details on the experimental set-up, see Appendix G. The code for reproducing all experiments is available at `https://github.com/marwash25/subpruning`.

**LeNet on MNIST**  We pre-train LeNet model on MNIST achieving $97.75\%$ top-1 accuracy. We prune all layers except the last classifier layer. Results are presented in Figure 1 (left). All three variants of our method consistently outperform other baselines, even when reweighting is applied to them, with ASYMINCHANGE doing the best and LAYERINCHANGE the worst. We observe that reweighting significantly improves the performance of all methods except LAYERGREEDYFS variants.

**ResNet56 on CIFAR-10**  We use the ResNet56 model pre-trained on CIFAR-10 provided in ShrinkBench [Blalock et al., 2020], which achieves $92.27\%$ top-1 accuracy. We prune all layers except the last layer in each residual branch, the last layer before each residual branch, and the last classifier layer. Results are presented in Figure 1 (middle). The sequential variants of our method perform the best. Their performance is closely matched by LAYERWEIGHTNORM and ACTGRAD (with reweighting) for most compression ratios, except very large ones. LAYERINCHANGE performs significantly worst here than the sequential variants of our method. This is likely due to the larger number of layers pruned in ResNet56 compared to LeNet (27 vs 4 layers), which increases the effect of pruning earlier layers on subsequent ones. Here also reweighting improves the performance of all methods except the LAYERGREEDYFS variants.

**VGG11 on CIFAR-10**  We pre-train VGG11 model on CIFAR-10 obtaining $90.11\%$ top-1 accuracy. We prune all layers except the last features layer and the last classifier layer. Results are presented in Figure 1 (right). The three variants of our method perform the best. Their performance is matched by ACTGRAD and LAYERWEIGHTNORM (with reweighting). LAYERINCHANGE performs similarly to the sequential variants of our method here, even slightly better at compression ratio 32, probably because the number of layers being pruned is again relatively small (9 layers). As before, reweighting benefits all methods except the LAYERGREEDYFS variants.

**Discussion**  We summarize our observations from the empirical results:

- Our proposed pruning method outperforms state-of-the-art structured pruning methods in various one shot pruning settings. As expected, ASYMINCHANGE is the best performing variant of our method, and LAYERINCHANGE the worst, with its performance deteriorating with deeper models. Our results also illustrate the robustness of our method, as it reliably yields the best results in various settings, while other baselines perform well in some settings but not in others.

- Reweighting significantly improves performance for all methods, except LAYERGREEDYFS and LAYERGREEDYFS-fd. We suspect that reweighting does not help in this case because this

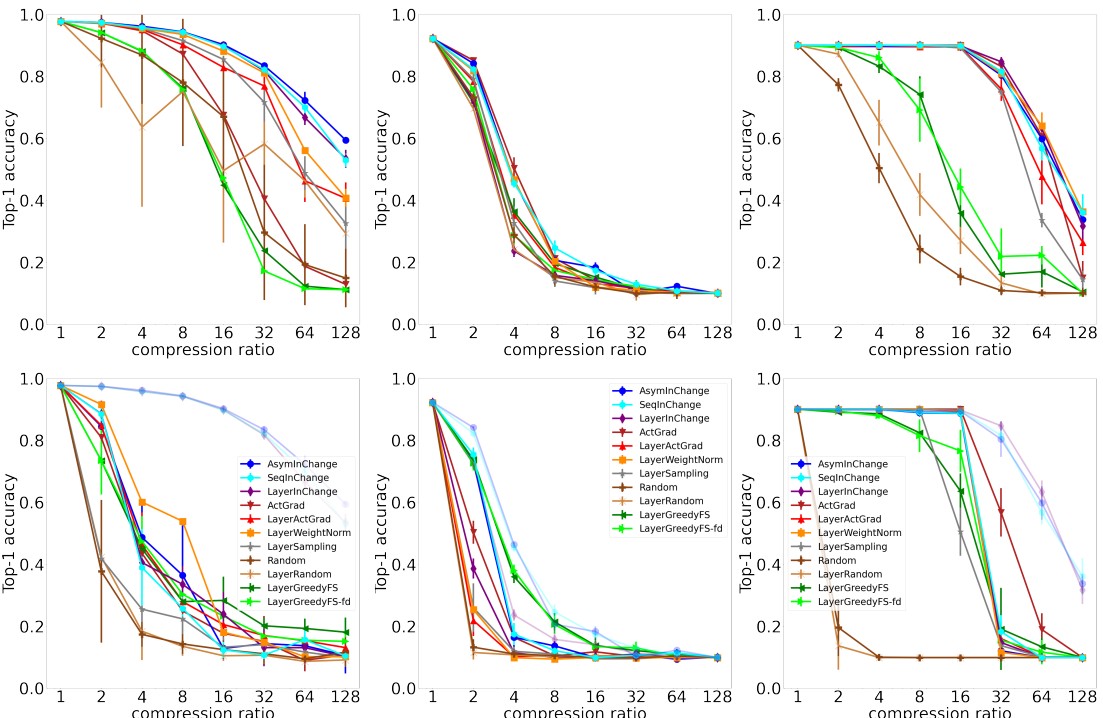

Figure 1: Top-1 Accuracy of different pruning methods applied to LeNet on MNIST (left), ResNet56 on CIFAR10 (middle), and VGG11 on CIFAR10 (right), for several compression ratios (in log-scale), with (top) and without (bottom) reweighting. We include the three reweighted variants of our method in the bottom plots (faded) for reference.

method already scales the next layer weights, and it takes into account this scaling when selecting neurons/channels to keep, so replacing it with reweighting can hurt performance.

- The choice of how much to prune in each layer given a global budget can have a drastic effect on performance, as illustrated in Appendix I.

- Fine-tuning with full-training data boosts performance more than reweighting, while fine-tuning with limited data helps less, as illustrated in Appendix H. Reweighting still helps when fine-tuning with limited-data, except for LAYERGREEDYFS variants, but it can actually deteriorate performance when fine-tuning with full-data. Our method still outperforms other baselines after fine-tuning with limited-data, and is among the best performing methods even in the full-data setting.

## 8 Conclusion

We proposed a data-efficient structured pruning method, based on submodular optimization. By casting the layerwise subset selection problem as a weakly submodular optimization problem, we are able to use the GREEDY algorithm to provably approximate it. Empirically, our method consistently outperforms existing structured pruning methods on different network architectures and datasets.

## Acknowledgments and Disclosure of Funding

We thank Stefanie Jegelka, Debadeepta Dey, Jose Gallego-Posada for their helpful discussions, and Yan Zhang, Boris Knyazev for their help with running experiments. We also acknowledge the MIT SuperCloud and Lincoln Laboratory Supercomputing Center (supercloud.mit.edu), Compute Canada (www.computecanada.ca), Calcul Quebec (www.calculquebec.ca), WestGrid (www.westgrid.ca), ACENET (ace-net.ca), the Mila IDT team, Idiap Research Institute and the Machine Learning Research Group at the University of Guelph, for providing HPC resources that have contributed to the research results reported within this paper. This research was partially supported by the Canada CIFAR AI Chair Program. Simon Lacoste-Julien is a CIFAR Associate Fellow in the Learning Machines & Brains program.

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
