Ye et al. [2020a] select neurons/channels to keep in a given layer that minimize the loss of the pruned network. More precisely, they are solving

$$\min_{\alpha \geq 0, \sum_i \alpha_i = 1} \sum_{i=1}^{n} [(f_\alpha^\ell(x_i) - y_i)^2],$$

where $(x_i, y_i)$ are data points, $f_\alpha^\ell(x)$ is the output of the model where neurons in layer $\ell$ corresponding to $\alpha_i = 0$ are pruned and the weights of the next layer are scaled by $\alpha_i n_\ell$, i.e., $A^\ell W^{\ell+1}$ is replaced by $n_\ell A^\ell \text{Diag}(\alpha) W^{\ell+1}$ where $\text{Diag}(\alpha)$ is the diagonal matrix with $\alpha$ as its diagonal. This is similar to the $\ell_1$-relaxation of the selection problem (1) we solve, in the special case of a two layer network with a single output, and instead of optimizing the weights of the next layer like we do, they optimize how much to scale them, i.e., in this case their selection problem reduces to

$$\min_{\alpha \geq 0, \sum_i \alpha_i = 1} \|A^\ell w^{\ell+1} - n_\ell A^\ell \text{Diag}(\alpha) w^{\ell+1}\|_2^2.$$

They use a greedy algorithm with Frank-Wolfe like updates to approximate it (see [Ye et al., 2020a, Section 12.1] for the relation between their greedy algorithm and Frank-Wolfe algorithm). This method is very expensive as it requires $O(kn_\ell)$ forward passes in the full network, to prune each layer. The provided theoretical guarantees only holds for two layer networks, and are with respect to an $\ell_1$-relaxation of the selection problem. Empirically, our method significantly outperforms the method of [Ye et al., 2020a] in all settings we consider.

Ye et al. [2020b] propose two pruning method: Greedy Global imitation and Greedy Local imitation. Greedy Global imitation is the same method from [Ye et al., 2020a] but with an additional approximation technique which reduces the cost of pruning one layer from $O(kn_\ell)$ to $O(k)$ forward passes through the full network. This is still more expensive than the cost of our method which is equivalent to $O(1/\epsilon)$ forward passes through only the layer being pruned, if using the fast Greedy algorithm from [Li et al., 2022] (see Section 3.3). Greedy Local imitation is closer to our approach, as it selects neurons/channels to keep in a given layer that minimize the change in the input to the next layer, but it also solves an $\ell_1$-relaxation of the selection problem and only optimize the scaling of the next layer weights instead of the weights directly, i.e., it solves

$$\min_{\alpha \geq 0, \sum_i \alpha_i = 1} \|A^\ell w^{\ell+1} - n_\ell A^\ell \text{Diag}(\alpha) w^{\ell+1}\|_2^2.$$

A similar greedy algorithm with Frank-Wolfe like updates as in [Ye et al., 2020a] is used. Although the selection problem solved is simpler than ours, the cost of pruning one layer is still more expensive than ours: $O(kn_\ell n_{\ell+1} n)$ vs $O((n_\ell)^2(n_{\ell+1} + n + k)/\epsilon)$. Ye et al. [2020b] also provide bounds on the difference between the output of the original network and the pruned one, with exponential convergence rate for Greedy Local imitation, and $O(1/k^2)$ rate for the Greedy Global imitation. Similar guarantees with exponential convergence rate hold for our method (see Section 6). Empirically, the results in [Ye et al., 2020b] show that their global method typically outperforms their local one. So we expect our method to also outperform their local method, since it outperforms their global method.

# B    Missing proofs

Recall that $F(S) = \|A^\ell W^{\ell+1}\|_F^2 - \min_{\tilde{W}^{\ell+1} \in \mathbb{R}^{n_\ell \times n_{\ell+1}}} \|A^\ell W^{\ell+1} - A_S^\ell \tilde{W}^{\ell+1}\|_F^2$, and $G(S) = F(M(S))$, where $M$ maps each channel to its corresponding columns in $A^\ell$. We denote by $\tilde{F}(S)$ the objective corresponding to the asymmetric formulation introduced in Section 5.1, i.e., $\tilde{F}(S) = \|A^\ell W^{\ell+1}\|_F^2 - \min_{\tilde{W}^{\ell+1} \in \mathbb{R}^{n_\ell \times n_{\ell+1}}} \|A^\ell W^{\ell+1} - B_S^\ell \tilde{W}^{\ell+1}\|_F^2$, and similarly $\tilde{G}(S) = \tilde{F}(M(S))$, where $M$ maps each channel to its corresponding columns in $A^\ell$.

We introduce some notation that will be used throughout the Appendix. Given any matrix $D$ and vector $y$, we denote by $x^S(y) \in \arg\min_{\text{supp}(x) \subseteq S} \frac{1}{2}\|y - Dx\|_2^2$ the vector of optimal regression coefficients, and by $\text{proj}_S(y) = Dx^S(y)$, $R^S(y) = y - \text{proj}_S(y)$ the corresponding projection and residual.

## B.1 Submodularity ratio bounds: Proof of Proposition 3.1 and 4.1 and their extension to the asymmetric formulation

In this section, we prove that $F, G$, and their asymmetric variants $\tilde{F}, \tilde{G}$ are all non-decreasing weakly submodular functions. We start by reviewing the definition of restricted smoothness (RSM) and restricted strong convexity(RSC).

**Definition B.1** (RSM/RSC). Given a differentiable function $\ell : \mathbb{R}^d \to \mathbb{R}$ and $\Omega \subset \mathbb{R}^d \times \mathbb{R}^d$, $\ell$ is $\mu_\Omega$-RSC and $\nu_\Omega$-RSM if $\frac{\mu_\Omega}{2} \|x - y\|_2^2 \leq \ell(y) - \ell(x) - \langle \nabla \ell(x), y - x \rangle \leq \frac{\nu_\Omega}{2} \|x - y\|_2^2, \quad \forall (x, y) \in \Omega.$

If $\ell$ is RSC/RSM on $\Omega = \{(x, y) : \|x\|_0 \leq k, \|y\|_0 \leq k, \|x - y\|_0 \leq k\}$, we denote by $\mu_k, \nu_k$ the corresponding RSC and RSM parameters.

**Proposition 3.1.** *Given $U \subseteq V, k \in \mathbb{N}_+$, $F$ is a normalized non-decreasing $\gamma_{U,k}$-weakly submodular function, with*

$$\gamma_{U,k} \geq \frac{\min_{\|z\|_2=1, \|z\|_0 \leq |U|+k} \|A^\ell z\|_2^2}{\max_{\|z\|_2=1, \|z\|_0 \leq |U|+1} \|A^\ell z\|_2^2}.$$

*Proof.* We can write $F(S) = \sum_{m=1}^{n_{\ell+1}} F_m(S) := \ell_m(0) - \min_{\text{supp}(\tilde{w}_m) \subseteq S} \ell_m(\tilde{w}_m)$, where $\ell_m(\tilde{w}_m) = \|A^\ell w_m^{\ell+1} - A^\ell \tilde{w}_m\|_2^2$. The function $F_m(S)$ is then $\gamma_{U,k}$-weakly submodular with $\gamma_{U,k} \geq \frac{\mu_{|U|+k}}{\nu_{|U|+1}}$ [Elenberg et al., 2016], where $\mu_{|U|+k}$ and $\nu_{|U|+1}$ are the RSC and RSM parameters of $\ell_m$, given by $\mu_{|U|+k} = \min_{\|z\|_2=1, \|z\|_0 \leq |U|+k} \|A^\ell z\|_2^2$, and $\nu_{|U|+1} = \max_{\|z\|_2=1, \|z\|_0 \leq |U|+1} \|A^\ell z\|_2^2$. It follows then that $F$ is also $\gamma_{U,k}$-weakly submodular. It is easy to check that $F$ is also normalized and non-decreasing. $\square$

**Proposition 4.1.** *Given $U \subseteq V_\ell, k \in \mathbb{N}_+$, $G$ is a normalized non-decreasing $\gamma_{U,k}$-weakly submodular function, with*

$$\gamma_{U,k} \geq \frac{\min_{\|z\|_2=1, \|z\|_0 \leq r_h r_w(|U|+k)} \|A^\ell z\|_2^2}{\max_{\|z\|_2=1, \|z\|_0 \leq r_h r_w(|U|+1)} \|A^\ell z\|_2^2}.$$

*Proof.* By definition, $G$ is $\gamma_{U,k}$-weakly submodular iff $F$ satisfies

$$\gamma_{U,k} F(M(S)|M(L)) \leq \sum_{i \in S} F(M(i)|M(L)),$$

for every two disjoint sets $L, S \subseteq V_\ell$, such that $L \subseteq U, |S| \leq k$. We extend the relation established in [Elenberg et al., 2016] between weak submodularity and RSC/RSM parameters to this case.

Let $S' = M(S), L' = M(L), I' = M(i)$, and $k' = r_h r_w k$. As before, we can write $G(S) = F(S') = \sum_{m=1}^{n_{\ell+1}} F_m(S') := \ell_m(0) - \min_{\text{supp}(\tilde{w}_m) \subseteq S'} \ell_m(\tilde{w}_m)$, where $\ell_m(\tilde{w}_m) = \|A^\ell w_m^{\ell+1} - A^\ell \tilde{w}_m\|_2^2$. We denote by $\mu_k$ and $\nu_k$ the RSC and RSM parameters of $\ell_m$, given by $\mu_k = \min_{\|z\|_2=1, \|z\|_0 \leq k} \|A^\ell z\|_2^2$, and $\nu_k = \max_{\|z\|_2=1, \|z\|_0 \leq k} \|A^\ell z\|_2^2$. To simplify notation, we use $x^S := x^S(A^\ell w_m^{\ell+1})$.

For every two disjoint sets $L, S \subseteq V_\ell$, such that $L \subseteq U, |S| \leq k$, we have:

$$\begin{aligned}
0 \leq F_m(S'|L') &= \ell_m(x^{L'}) - \ell_m(x^{S' \cup L'}) \\
&\leq -\langle \nabla \ell_m(x^{L'}), x^{S' \cup L'} - x^{L'} \rangle - \frac{\mu_{|L'|+k'}}{2} \|x^{S' \cup L'} - x^{L'}\|_2^2 \\
&\leq \max_{\text{supp}(x) \subseteq S' \cup L'} -\langle \nabla \ell_m(x^{L'}), x - x^{L'} \rangle - \frac{\mu_{|L'|+k'}}{2} \|x - x^{L'}\|_2^2
\end{aligned}$$

By setting $x = x^{L'} - \frac{[\nabla \ell_m(x^{L'})]_{S'}}{\mu_{|L'|+k'}}$, we get $G(S'|L') \leq \frac{\|[\nabla \ell_m(x^{L'})]_{S'}\|_2^2}{2\mu_{|L'|+k'}}$.

Given any $i \in S$, $I' = M(i)$, we have

$$
\begin{aligned}
F_m(I'|L') &= \ell_m(x^{L'}) - \ell_m(x^{I' \cup L'}) \\
&\geq \ell_m(x^{L'}) - \ell_m(x^{L'} - \frac{[\nabla \ell_m(x^{L'})]_{I'}}{\nu_{|L'|+|I'|}}) \\
&\geq \langle \nabla \ell_m(x^{L'}), \frac{[\nabla \ell_m(x^{L'})]_I}{\nu_{|L'|+|I'|}} \rangle - \frac{\nu_{|L'|+k'}}{2} \|\frac{[\nabla \ell_m(x^{L'})]_I}{\nu_{|L'|+|I'|}}\|_2^2 \\
&= \frac{\|[\nabla \ell_m(x^{L'})]_{I'}\|_2^2}{2\nu_{|L'|+|I'|}}
\end{aligned}
$$

Hence,

$$
G(S|L) \leq \sum_{m=1}^{n_{\ell+1}} \frac{\|[\nabla \ell_m(x^{L'})]_{S'}\|_2^2}{2\mu_{|L'|+k'}} = \sum_{i \in S, I'=M(i)} \frac{\|[\nabla \ell_m(x^{L'})]_{I'}\|_2^2}{2\mu_{|L'|+k'}}
$$

$$
= \sum_{i \in S, I'=M(i)} \frac{\nu_{|L'|+|I'|}}{\mu_{|L'|+k'}} \frac{\|[\nabla \ell_m(x^{L'})]_{I'}\|_2^2}{2\nu_{|L'|+|I'|}} = \frac{\nu_{|L'|+k'}}{\mu_{|L'|+k'}} \sum_{i \in S} G(i|L).
$$

We thus have $\gamma_{U,k} \geq \frac{\mu_{|U'|+k'}}{\nu_{|U'|+|I'|}}$. $\square$

Both Proposition 3.1 and Proposition 4.1 apply also to the asymmetric variants, using exactly the same proofs.

**Proposition B.2.** *Given $U \subseteq V, k \in \mathbb{N}_+$, $\tilde{F}$ is a normalized non-decreasing $\gamma_{U,k}$-weakly submodular function, with*

$$
\gamma_{U,k} \geq \frac{\min_{\|z\|_2=1, \|z\|_0 \leq |U|+k} \|B^\ell z\|_2^2}{\max_{\|z\|_2=1, \|z\|_0 \leq |U|+1} \|B^\ell z\|_2^2}.
$$

**Proposition B.3.** *Given $U \subseteq V_\ell, k \in \mathbb{N}_+$, $G$ is a normalized non-decreasing $\gamma_{U,k}$-weakly submodular function, with*

$$
\gamma_{U,k} \geq \frac{\min_{\|z\|_2=1, \|z\|_0 \leq r_h r_w (|U|+k)} \|A^\ell z\|_2^2}{\max_{\|z\|_2=1, \|z\|_0 \leq r_h r_w (|U|+1)} \|A^\ell z\|_2^2}.
$$

## B.2 Cost bound: Proof of Proposition 3.2 and its extension to other variants

In this section, we investigate the cost of applying GREEDY with $F, G$ and their asymmetric variants $\tilde{F}, \tilde{G}$. To that end, we need the following key lemmas showing how to update the least squares solutions and the function values after adding one or more elements.

**Lemma B.4.** *Given a matrix $D$, vector $y$, and a vector of optimal regression coefficients $x^S(y) \in \arg\min_{\mathrm{supp}(x) \subseteq S} \frac{1}{2}\|y - Dx\|_2^2$, we have for all $S \subseteq V, i \notin S$:*

$$
x^{S \cup i}(y) = (x^S(y) - x^S(d_i)\gamma^{S,i}(y)) + \gamma^{S,i}(y)\mathbf{1}_i \in \arg\min_{\mathrm{supp}(x) \subseteq S \cup i} \frac{1}{2}\|y - Dx\|_2^2,
$$

*where $\gamma^{S,i}(y) \in \arg\min_{\gamma \in \mathbb{R}} \frac{1}{2}\|y - R^S(d_i)\gamma\|_2^2$. Hence, $\mathrm{proj}_{S \cup i}(y) = \mathrm{proj}_S(y) + \mathrm{proj}_{R^S(d_i)}(y)$, where $\mathrm{proj}_{R^S(d_i)}(y) = R^S(d_i)\gamma^{S,i}(y)$.*

*Similarly, for $I \subseteq V \setminus S$, let $R^S(D_I)$ be the matrix with columns $R^S(d_i)$, $x^S(D_I)$ the matrix with columns $x^S(d_i)$, and $\gamma^{S,I}(y) \in \arg\min_{\gamma \in \mathbb{R}^{|I|}} \frac{1}{2}\|y - R^S(D_I)\gamma\|_2^2$, then*

$$
x^{S \cup I}(y) = (x^S(y) - x^S(D_I)\gamma^{S,I}(y)) + e_I \gamma^{S,I}(y) \in \arg\min_{\mathrm{supp}(x) \subseteq S \cup I} \frac{1}{2}\|y - Dx\|_2^2,
$$

*where $e_I \in \mathbb{R}^{|V| \times |I|}$ is the matrix with $[e_I]_{i,i} = 1$ for all $i \in I$, and $0$ elsewhere. Hence, $\mathrm{proj}_{S \cup I}(y) = \mathrm{proj}_S(y) + \mathrm{proj}_{R^S(D_I)}(y)$, where $\mathrm{proj}_{R^S(D_I)}(y) = R^S(D_I)\gamma^{S,I}(y)$.*

*Proof.* By optimality conditions, we have:

$$D_S^\top (D_S x^S(y) - y) = 0 \tag{7}$$

$$D_S^\top (D_S x^S(d_i) - d_i) = 0 \Rightarrow -D_S^\top R^S(d_i) = 0 \tag{8}$$

$$R^S(d_i)^\top (R^S(d_i)\gamma^{S,i}(y) - y) = 0 \tag{9}$$

We prove that $\hat{x}^{S\cup i}(y) = (x^S(y) - x^S(d_i)\gamma^{S,i}(y)) + \gamma^{S,i}(y)\mathbf{1}_i$ satisfies the optimality conditions on $x^{S\cup i}(y)$, hence $\hat{x}^{S\cup i}(y) = x^{S\cup i}(y)$.

We have $D_{S\cup i}\hat{x}^{S\cup i}(y) = D_S x^S(y) + R^S(d_i)\gamma^{S,i}(y)$, then

$$D_S^\top(D_{S\cup i}\hat{x}^{S\cup i}(y) - y) = D_S^\top(D_S x^S(y) - y) + D_S^\top R^S(d_i)\gamma^{S,i}(y) = 0$$

and

$$
\begin{aligned}
d_i^\top(D_{S\cup i}\hat{x}^{S\cup i}(y) - y) &= (R^S(d_i) + D_S x^S(d_i))^\top(D_S x^S(y) + R^S(d_i)\gamma^{S,i}(y) - y) \\
&= R^S(d_i)^\top D_S x^S(y) + R^S(d_i)^\top (R^S(d_i)\gamma^{S,i}(y) - y) \\
&\quad + (D_S x^S(d_i))^\top(D_S x^S(y) - y) + (D_S x^S(d_i))^\top R^S(d_i)\gamma^{S,i}(y) \\
&= 0
\end{aligned}
$$

The proof for the case where we add multiple indices at once follows similarly. $\square$

**Lemma B.5.** *For all $S \subseteq V_\ell, i \notin S$, let $R_S(b_i^\ell) = b_i^\ell - \mathrm{proj}_S(b_i^\ell)$, $\mathrm{proj}_{R_S(b_i^\ell)}(y) = R_S(b_i^\ell)\gamma^{S,i}(y)$ with $\gamma^{S,i}(y) \in \arg\min_{\gamma\in\mathbb{R}} \|y - R_S(b_i^\ell)\gamma\|_2^2$. We can write the marginal gain of adding $i$ to $S$ w.r.t $\tilde{F}$ as:*

$$\tilde{F}(i|S) = \sum_{m=1}^{n_{\ell+1}} \|\mathrm{proj}_{R_S(b_i^\ell)}(A^\ell)w_m^{\ell+1}\|_2^2,$$

*where $\mathrm{proj}_{R_S(b_i^\ell)}(A^\ell)$ is the matrix with columns $\mathrm{proj}_{R_S(b_i^\ell)}(a_j^\ell)$ for all $j \in V_\ell$. Similarly, for all $S \subseteq V_\ell, I \subseteq V_\ell \setminus S$, let $R_S(B_I^\ell) = B_I^\ell - \mathrm{proj}_S(B_I^\ell)$, $\mathrm{proj}_{R_S(B_I^\ell)}(y) = R_S(B_I^\ell)\gamma^{S,I}(y)$ with $\gamma^{S,I}(y) \in \arg\min_{\gamma\in\mathbb{R}^{|I|}} \|y - R_S(B_I^\ell)\gamma\|_2^2$. We can write the marginal gain of adding $I$ to $S$ w.r.t $\tilde{F}$ as:*

$$\tilde{F}(I|S) = \sum_{m=1}^{n_{\ell+1}} \|\mathrm{proj}_{R_S(B_I^\ell)}(A^\ell)w_m^{\ell+1}\|_2^2,$$

*where $\mathrm{proj}_{R_S(B_I^\ell)}(A^\ell)$ is the matrix with columns $\mathrm{proj}_{R_S(B_I^\ell)}(a_j^\ell)$ for all $j \in V_\ell$.*

*Proof.* We prove the claim for the case where we add several elements. The case where we add a single element then follows as a special case. For a fixed $S \subseteq V_\ell$, the reweighted asymmetric input change $\|A^\ell W^{\ell+1} - B_S^\ell \tilde{W}^{\ell+1}\|_F^2$ is minimized by setting $\tilde{W}^{\ell+1} = x^S(A^\ell)W^{\ell+1}$, where $x^S(A^\ell) \in \mathbb{R}^{n_\ell \times n_\ell}$ is the matrix with columns $x^S(a_j^\ell)$ such that

$$x^S(a_j^\ell) \in \arg\min_{\mathrm{supp}(x)\subseteq S} \|a_j^\ell - B^\ell x\|_2^2 \text{ for all } j \in V_\ell. \tag{10}$$

Plugging $\tilde{W}^{\ell+1}$ into the expression of $\tilde{F}(S)$ yields $\tilde{F}(S) = \|A^\ell W^{\ell+1}\|_F^2 - \|(A^\ell - \mathrm{proj}_S(A^\ell))W^{\ell+1}\|_F^2$, where $\mathrm{proj}_S(A^\ell) = B_S^\ell x^S(A^\ell)$. For every $m \in \{1, \cdots, n_{\ell+1}\}$, we have:

$$\|(\mathrm{proj}_{S\cup I}(A^\ell) - A^\ell)w_m^{\ell+1}\|_2^2 - \|(\mathrm{proj}_S(A^\ell) - A^\ell)w_m^{\ell+1}\|_2^2$$

$$= \|(\mathrm{proj}_S(A^\ell) + \mathrm{proj}_{R_S(B_I)}(A^\ell) - A^\ell)w_m^{\ell+1}\|_2^2 - \|(\mathrm{proj}_S(A^\ell) - A^\ell)w_m^{\ell+1}\|_2^2 \quad \text{(by Lemma B.4)}$$

$$= \|\mathrm{proj}_{R_S(B_I)}(A^\ell)w_m^{\ell+1}\|_2^2 - 2\langle(A^\ell - \mathrm{proj}_S(A^\ell))w_m^{\ell+1}, \mathrm{proj}_{R_S(B_I)}(A^\ell)w_m^{\ell+1}\rangle$$

$$= \|\mathrm{proj}_{R_S(B_I)}(A^\ell)w_m^{\ell+1}\|_2^2 - 2\langle\mathrm{proj}_{R_S(B_I)}(A^\ell)w_m^{\ell+1}, \mathrm{proj}_{R_S(B_I)}(A^\ell)w_m^{\ell+1}\rangle$$

$$= -\|\mathrm{proj}_{R_S(B_I)}(A^\ell)w_m^{\ell+1}\|_2^2$$

where the second to last equality holds because $y - \mathrm{proj}_S(y) - \mathrm{proj}_{R_S(B_I)}(y)$ and $\mathrm{proj}_{R_S(B_I)}(y')$ are orthogonal by optimality conditions (see proof of Lemma B.4):

$$\langle y - \mathrm{proj}_S(y) - \mathrm{proj}_{R_S(B_I)}(y), \mathrm{proj}_{R_S(B_I)}(y')\rangle = \langle y - B_S x^S(y) - R_S(B_I)\gamma^{S,I}(y), R_S(B_I)\gamma^{S,I}(y')\rangle = 0.$$

Hence, $\tilde{F}(I|S) = \sum_{m=1}^{n_{\ell+1}} \|\mathrm{proj}_{R_S(B_I)}(A^\ell)w_m^{\ell+1}\|_2^2$. In particular, if $I = \{i\}$, $\tilde{F}(i|S) = \sum_{m=1}^{n_{\ell+1}} \|\mathrm{proj}_{R_S(b_i^\ell)}(A^\ell)w_m^{\ell+1}\|_2^2$. $\square$

**Lemma B.6.** *For all $S \subseteq V_\ell, i \notin S$, let $R_S(a_i^\ell) = a_i^\ell - \text{proj}_S(a_i^\ell)$, $\text{proj}_{R_S(a_i^\ell)}(y) = R_S(a_i^\ell)\gamma^{S,i}(y)$ with $\gamma^{S,i}(y) \in \arg\min_{\gamma \in \mathbb{R}} \|y - R_S(a_i^\ell)\gamma\|_2^2$. We can write the marginal gain of adding $i$ to $S$ w.r.t $F$ as:*

$$F(i|S) = \sum_{m=1}^{n_{\ell+1}} \|\text{proj}_{R_S(a_i^\ell)}(A_{V\setminus S}^\ell)w_m^{\ell+1}\|_2^2,$$

*where $\text{proj}_{R_S(a_i^\ell)}(A_{V\setminus S}^\ell)$ is the matrix with columns $\text{proj}_{R_S(a_i^\ell)}(a_j^\ell)$ for all $j \in V \setminus S$, 0 otherwise. Similarly, for all $S \subseteq V_\ell, I \subseteq V_\ell \setminus S$, let $R_S(A_I^\ell) = A_I^\ell - \text{proj}_S(A_I^\ell)$, $\text{proj}_{R_S(A_I^\ell)}(y) = R_S(A_I^\ell)\gamma^{S,I}(y)$ with $\gamma^{S,I}(y) \in \arg\min_{\gamma \in \mathbb{R}^{|I|}} \|y - R_S(A_I^\ell)\gamma\|_2^2$. We can write the marginal gain of adding $I$ to $S$ w.r.t $F$ as:*

$$F(I|S) = \sum_{m=1}^{n_{\ell+1}} \|\text{proj}_{R_S(A_I^\ell)}(A_{V\setminus S}^\ell)w_m^{\ell+1}\|_2^2,$$

*where $\text{proj}_{R_S(A_I^\ell)}(A_{V\setminus S}^\ell)$ is the matrix with columns $\text{proj}_{R_S(A_I^\ell)}(a_j^\ell)$ for all $j \in V \setminus S$, 0 otherwise.*

*Proof.* Setting $B^\ell = A^\ell$ in Lemma B.5, we get $F(I|S) = \sum_{m=1}^{n_{\ell+1}} \|\text{proj}_{R_S(A_I^\ell)}(A^\ell)w_m^{\ell+1}\|_2^2$. Note that for all $i \in I, j \in S$, $a_j^\ell$ and $R_S(a_i^\ell)$ are orthogonal, and hence $\text{proj}_{R_S(A_I^\ell)}(a_j^\ell) = 0$, by optimality conditions (see proof of Lemma B.4). It follows then that $F(I|S) = \sum_{m=1}^{n_{\ell+1}} \|\text{proj}_{R_S(A_I^\ell)}(A_{V\setminus S}^\ell)w_m^{\ell+1}\|_2^2$. In particular, if $I = \{i\}$, $F(i|S) = \sum_{m=1}^{n_{\ell+1}} \|\text{proj}_{R_S(a_i^\ell)}(A_{V\setminus S}^\ell)w_m^{\ell+1}\|_2^2$. $\square$

**Proposition B.7.** *Given $S \subseteq V_\ell$ such that $|S| \le k$, $i \notin S$, let $\text{proj}_S(a_j^\ell) = A_S^\ell x^S(a_j^\ell)$. Assuming $\tilde{F}(S), \text{proj}_S(b_j^\ell), x^S(b_j^\ell)$ for all $j \notin S$, and $x^S(a_j^\ell)$ for all $j \in V_\ell$, were computed in the previous iteration, we can compute $\tilde{F}(S+i), \text{proj}_{S+i}(b_j^\ell), x^{S+i}(b_j^\ell)$ for all $j \notin (S+i)$, and $x^{S+i}(a_j^\ell)$ for all $j \in V_\ell$, in*

$$O(n_\ell \cdot (n_{\ell+1} + n + k)) \text{ time.}$$

*Computing the optimal weights in Eq. (4) at the end of GREEDY can then be done in $O(k \cdot n_\ell \cdot n_{\ell+1})$ time.*

*Proof.* By Lemma B.5, we can update the function value using

$$\tilde{F}(i|S) = \sum_{m=1}^{n_{\ell+1}} \|\text{proj}_{R_S(b_i^\ell)}(A^\ell)w_m^{\ell+1}\|_2^2 = \sum_{m=1}^{n_{\ell+1}} \Big(\sum_{j \in V_\ell} \gamma^{S,i}(a_j^\ell)w_{jm}^{\ell+1}\Big)^2 \|R_S(b_i^\ell)\|_2^2.$$

This requires $O(n)$ to compute $R_S(b_i^\ell)$ and its norm, $O(n_\ell \cdot n)$ to compute $\gamma^{S,i}(a_j^\ell) = \frac{R_S(b_i^\ell)^\top a_j^\ell}{\|R_S(b_i^\ell)\|_2^2}$ for all $j \in V_\ell$, and an additional $O(n_\ell \cdot n_{\ell+1})$ to finally evaluate $\tilde{F}(S+i)$. We also need $O(|S^c| \cdot n)$ to update $\text{proj}_{S\cup i}(b_j) = \text{proj}_S(b_j) + \text{proj}_{R_S(b_i^\ell)}(b_j)$ (by Lemma B.4), using $\text{proj}_{R_S(b_i^\ell)}(b_j) = R_S(b_i^\ell)\gamma^{S,i}(b_j^\ell)$ for all $j \in V_\ell \setminus S \cup i$, $O(|S^c| \cdot |S|)$ to update $x^{S\cup i}(b_j^\ell) = x^S(b_j^\ell) + (\mathbf{1}_i - x^S(b_i))\gamma^{S,i}(b_j^\ell))$ (by Lemma B.4) for all $j \in V_\ell \setminus S \cup i$, and $O(n_\ell \cdot |S|)$ to update $x^{S\cup i}(a_j^\ell) = x^S(a_j^\ell) + (\mathbf{1}_i - x^S(b_i))\gamma^{S,i}(a_j^\ell))$ for all $j \in V_\ell$. So in total, we need $O(n_\ell \cdot (n_{\ell+1} + n + k))$. Computing the new weights $\tilde{W}^{\ell+1} = x^S(A^\ell)W^{\ell+1}$ at the end can be done in $O(n_\ell \cdot n_{\ell+1} \cdot |S|) = O(n_\ell \cdot n_{\ell+1} \cdot k)$. $\square$

**Proposition 3.2.** *Given $S \subseteq V_\ell$ such that $|S| \le k$, $i \notin S$, let $\text{proj}_S(a_j^\ell) = A_S^\ell x^S(a_j^\ell)$ be the projection of $a_j^\ell$ onto the column space of $A_S^\ell$, $R_S(a_i^\ell) = a_i^\ell - \text{proj}_S(a_i^\ell)$ and $\text{proj}_{R_S(a_i^\ell)}(a_j^\ell) \in \arg\min_{z=R_S(a_i^\ell)\gamma,\gamma\in\mathbb{R}} \|a_j^\ell - z\|_2^2$ the corresponding residual and the projection of $a_j^\ell$ onto it. We can write*

$$F(i|S) = \sum_{m=1}^{n_{\ell+1}} \|\text{proj}_{R_S(a_i^\ell)}(A_{V\setminus S}^\ell)w_m^{\ell+1}\|_2^2,$$

*where* $\text{proj}_{R_S(a_i)}(A^\ell_{V \setminus S})$ *is the matrix with columns* $\text{proj}_{R_S(a_i)}(a^\ell_j)$ *for all* $j \notin S$, $0$ *otherwise.*
*Assuming* $F(S), \text{proj}_S(a^\ell_j)$ *and* $x^S(a^\ell_j)$ *for all* $j \notin S$ *were computed in the previous iteration, we can compute* $F(S+i), \text{proj}_{S+i}(a^\ell_j)$ *and* $x^{S+i}(a^\ell_j)$ *for all* $j \notin (S+i)$ *in*

$$O(n_\ell \cdot (n_{\ell+1} + n + k)) \text{ time.}$$

*The optimal weights in Eq.* (4) *can then be computed in* $O(k \cdot n_\ell \cdot n_{\ell+1})$ *time, at the end of* GREEDY.

*Proof.* The proof follows from Lemma B.6 and B.4 in the same way as in Proposition B.7. $\quad\square$

Proposition 3.2 and Proposition B.7 apply also to $G$ and $\tilde{G}$ respectively, since $|M(i)| = O(1)$.

### B.3 Extension of STOCHASTIC-GREEDY to weakly submodular functions

In this section, we show that the guarantee of STOCHASTIC-GREEDY (Algorithm 2) easily extends to weakly submodular functions.

---

**Algorithm 2** STOCHASTIC-GREEDY

---

1: **Input:** Ground set $V$, set function $F : 2^V \to \mathbb{R}_+$, budget $k \in \mathbb{N}_+$
2: $S \leftarrow \emptyset$
3: **while** $|S| < k$ **do**
4: $\quad R \leftarrow$ a random subset obtained by sampling $s$ random elements from $V \setminus S$.
5: $\quad i^* \leftarrow \arg\max_{i \in R} F(i \mid S)$
6: $\quad S \leftarrow S \cup \{i^*\}$
7: **end while**
8: **Output:** $S$

---

**Proposition B.8.** *Let* $\hat{S}$ *be the solution returned by* STOCHASTIC-GREEDY *with* $s = \frac{n}{k}\log(\frac{1}{\epsilon})$, *and let* $F$ *be a non-negative monotone* $\gamma_{\hat{S},k}$-*weakly submodular function. Then*

$$\mathbb{E}[F(\hat{S})] \geq (1 - e^{-\gamma_{\hat{S},k}}) \max_{|S| \leq k} F(S)$$

*Proof.* Denote by $S_t$ the solution at iteration $t$ of STOCHASTIC-GREEDY, and $S^*$ an optimal solution. The proof follows in the same way as in [Mirzasoleiman et al., 2015, Theorem 1]. In particular, Lemma 2 therein, does not use submodularity, so it holds here too. It states that the expected gain of STOCHASTIC-GREEDY in one step is at least $\frac{1-\epsilon}{k} \sum_{i \in S^* \setminus S_t} F(i|S_t)$ for any $t$. Therefore,

$$\mathbb{E}[F(S_{t+1}) - F(S_t) \mid S_t] \geq \frac{1 - \epsilon}{k} \sum_{i \in S^* \setminus S_t} F(i|S_t)$$

$$\geq \gamma_{\hat{S},k} \frac{1 - \epsilon}{k} F(S^* \setminus S_t | S_t)$$

$$\geq \gamma_{\hat{S},k} \frac{1 - \epsilon}{k} (F(S^*) - F(S_t))$$

By taking expectation over $S_t$ and induction, we get

$$\mathbb{E}[F(S_t)] \geq \left(1 - \left(1 - \gamma_{\hat{S},k} \frac{1 - \epsilon}{k}\right)^k\right) F(S^*)$$

$$\geq \left(1 - e^{-\gamma_{\hat{S},k}(1-\epsilon)}\right) F(S^*)$$

$$\geq \left(1 - e^{-\gamma_{\hat{S},k}} - \epsilon\right) F(S^*)$$

$$\square$$

# C Error rates: Proof of Proposition 6.1 and 6.3

In this section, we provide the proofs of our method's error rates.

**Proposition 3.1.** *Given $U \subseteq V, k \in \mathbb{N}_+$, $F$ is a normalized non-decreasing $\gamma_{U,k}$-weakly submodular function, with*

$$\gamma_{U,k} \geq \frac{\min_{\|z\|_2=1, \|z\|_0 \leq |U|+k} \|A^\ell z\|_2^2}{\max_{\|z\|_2=1, \|z\|_0 \leq |U|+1} \|A^\ell z\|_2^2}.$$

*Proof.* This follows by extending the approximation guarantee in Eq. (3) to:

$$F(\hat{S}) \geq (1 - e^{-\gamma_{\hat{S},k^*} k/k^*}) \max_{|S| \leq k^*} F(S),$$

by a slight adaption of the proof in [Elenberg et al., 2016, Das and Kempe, 2011]. In particular, taking $k^* = n_\ell$ yields:

$$F(\hat{S}) = \|A^\ell W^{\ell+1}\|_F^2 - \|A^\ell W^{\ell+1} - A_{\hat{S}}^\ell \hat{W}^{\ell+1}\|_F^2 \leq (1 - e^{-\gamma_{\hat{S},n_\ell} k/n_\ell}) \|A^\ell W^{\ell+1}\|_F^2.$$

The first part of the claim follows by rearraging terms. Similarly, for the asymmetric formulation we have:

$$\tilde{F}(\hat{S}) \geq (1 - e^{-\gamma_{\hat{S},k^*} k/k^*}) \max_{|S| \leq k^*} \tilde{F}(S).$$

Taking $k^* = n_\ell$ yields:

$$\tilde{F}(\hat{S}) = \|A^\ell W^{\ell+1}\|_F^2 - \|A^\ell W^{\ell+1} - B_{\hat{S}}^\ell \hat{W}^{\ell+1}\|_F^2$$
$$\geq (1 - e^{-\gamma_{\hat{S},n_\ell} k/n_\ell})(\|A^\ell W^{\ell+1}\|_F^2 - \min_{\tilde{W}^{\ell+1} \in \mathbb{R}^{n_\ell \times n_{\ell+1}}} \|A^\ell W^{\ell+1} - B^\ell \tilde{W}^{\ell+1}\|_F^2)$$

The second part of the claim follows by rearraging terms. $\square$

**Corollary 6.3.** *Let $y \in \mathbb{R}^n$ be the original model output, $y^{\hat{S}_\ell}, y^{\tilde{S}_\ell} \in \mathbb{R}^n$ the outputs after layers 1 to $\ell$ are sequentially pruned using* SEQINCHANGE *and* ASYMINCHANGE, *respectively, and $H_\ell$ the function corresponding to all (unpruned) layers coming after layer $\ell$. If every function $H_\ell$ is Lipschitz continuous with constant $\|H_\ell\|_{Lip}$, then*

$$\|y - y^{\hat{S}_L}\|_2^2 \leq \sum_{\ell=1}^{L} e^{-\gamma_{\hat{S}_\ell, n_\ell} k_\ell/n_\ell} \|H_\ell\|_{Lip}^2 \|A^\ell W^{\ell+1}\|_F^2,$$

*and*

$$\|y - y^{\tilde{S}_L}\|_2^2 \leq \sum_{\ell=1}^{L} \prod_{\ell'=\ell+1}^{L} (1 - e^{-\gamma_{\tilde{S}_{\ell'}, n_{\ell'}} k_{\ell'}/n_{\ell'}}) e^{-\gamma_{\tilde{S}_\ell, n_\ell} k_\ell/n_\ell} \|H_\ell\|_{Lip}^2 \|A^\ell W^{\ell+1}\|_F^2.$$

*Proof.* We start by proving the bound for SEQINCHANGE. Recall that $B^\ell$ are the updated activations of layer $\ell$ after layers 1 to $\ell-1$ are pruned, and let $\hat{W}^{\ell+1}$ be the optimal weights corresponding to $\hat{S}_\ell$ (Eq. (4)). We can write $y = H_1(A^1 W^2) = H_1(B^1 W^2)$ since $A^1 = B^1$, and $y^{\hat{S}_\ell} = H_\ell(B_{\hat{S}_\ell}^\ell \hat{W}^{\ell+1}), y^{\hat{S}_{\ell-1}} = H_\ell(B^\ell W^{\ell+1})$ for all $\ell \in [L]$. Since $H_\ell$ is Lipschitz continuous for all $\ell \in [L]$, we have by the triangle inequality:

$$\|y - y^{\hat{S}_L}\|_2^2 \leq \sum_{\ell=1}^{L} \|y^{\hat{S}_\ell} - y^{\hat{S}_{\ell-1}}\|_F^2$$

$$\leq \sum_{\ell=1}^{L} \|H_\ell(B_{\hat{S}_\ell}^\ell \hat{W}^{\ell+1}) - H_\ell(B^\ell W^{\ell+1})\|_F^2$$

$$\leq \sum_{\ell=1}^{L} \|H_\ell\|_{Lip}^2 \|B_{\hat{S}_\ell}^\ell \hat{W}^{\ell+1} - B^\ell W^{\ell+1}\|_F^2$$

$$\leq \sum_{\ell=1}^{L} e^{-\gamma_{\hat{S}_\ell, n_\ell} k_\ell/n_\ell} \|H_\ell\|_{Lip}^2 \|B^\ell W^{\ell+1}\|_F^2,$$

where the last inequality follows from Proposition 6.1.

Next, we prove the bound for ASYMINCHANGE. Let $\tilde{W}^{\ell+1}$ be the optimal weights corresponding to $\tilde{S}_\ell$ (Eq. (4)). We can write $y = H_\ell(A^\ell W^{\ell+1})$ and $y^{\tilde{S}_\ell} = H_\ell(B^\ell_{\tilde{S}_\ell} \tilde{W}^{\ell+1})$ for all $\ell \in [L]$. By Proposition 6.1 and the Lipschitz continuity of $H_L$, we have

$$\|y - y^{\tilde{S}_L}\|_2^2 \le e^{-\gamma_{\tilde{S}_L, n_L} k_L/n_L} \|H_L\|_{\text{Lip}}^2 \|A^L W^{L+1}\|_F^2 + (1 - e^{-\gamma_{\tilde{S}_L, n_L} k_L/n_L}) \|H_L\|_{\text{Lip}}^2 \|A^L W^{L+1} - B^L W^{L+1}\|_F^2$$

Let $H_\ell^{\ell+1}$ denote the function corresponding to all layers between the $\ell$th layer and the $(\ell+1)$th layer (not necessarily consecutive in the network), then we can write $A^L W^{L+1} = H_{L-1}^L(A^{L-1} W^L)$ and $B^L W^{L+1} = H_{L-1}^L(B^{L-1}_{\tilde{S}_{L-1}} \tilde{W}^L)$, and $H_{L-1}(Z) = H_L(H_{L-1}^L(Z))$. It follows then that $\|H_{L-1}\|_{\text{Lip}} = \|H_L\|_{\text{Lip}} \|H_{L-1}^L\|_{\text{Lip}}$, and

$$\|y - y^{\tilde{S}_L}\|_2^2 \le e^{-\gamma_{\tilde{S}_L, n_L} k_L/n_L} \|H_L\|_{\text{Lip}}^2 \|A^L W^{L+1}\|_F^2 + (1 - e^{-\gamma_{\tilde{S}_L, n_L} k_L/n_L}) \|H_L\|_{\text{Lip}}^2 \|H_{L-1}^L\|_{\text{Lip}}^2 \|A^{L-1} W^L - B^{L-1}_{\tilde{S}_{L-1}} \tilde{W}^L\|_F^2$$

$$\le e^{-\gamma_{\tilde{S}_L, n_L} k_L/n_L} \|H_L\|_{\text{Lip}}^2 \|A^L W^{L+1}\|_F^2 + (1 - e^{-\gamma_{\tilde{S}_L, n_L} k_L/n_L}) \|H_{L-1}\|_{\text{Lip}}^2 \|A^{L-1} W^L - B^{L-1}_{\tilde{S}_{L-1}} \tilde{W}^L\|_F^2$$

Repeatedly applying the same arguments yields the claim. □

## D  Stronger notion of approximate submodularity

In this section, we show that $F$ and $\tilde{F}$ satisfy stronger properties than the weak submodularity discussed in Section 3.1, which lead to a stronger approximation guarantee for GREEDY. These properties do not necessarily hold for $G$ and $\tilde{G}$.

### D.1  Additional preliminaries

We start by reviewing some preliminaries. Recall that a set function $F$ is *submodular* if it has diminishing marginal gains: $F(i \mid S) \ge F(i \mid T)$ for all $S \subseteq T, i \in V \setminus T$. If $-F$ is submodular, then $F$ is said to be *supermodular*, i.e., $F$ satisfies $F(i \mid S) \le F(i \mid T)$, for all $S \subseteq T, i \in V \setminus T$. When $F$ is both submodular and supermodular, it is said to be *modular*.

Relaxed notions of submodularity/supermodularity, called *weak DR-submodularity/supermodularity*, were introduced in [Lehmann et al., 2006] and [Bian et al., 2017], respectively.

**Definition D.1** (Weak DR-sub/supermodularity). A set function $F$ is $\alpha_k$-weakly DR-submodular, with $k \in \mathbb{N}_+, \alpha_k > 0$, if

$$F(i|S) \ge \alpha_k F(i|T), \text{ for all } S \subseteq T, i \in V \setminus T, |T| \le k.$$

Similarly, $F$ is $\beta_k$-weakly DR-supermodular, with $k \in \mathbb{N}_+, \beta > 0$, if

$$F(i|T) \ge \beta_k F(i|S), \text{ for all } S \subseteq T, i \in V \setminus T, |T| \le k.$$

We say that $F$ is $(\alpha_k, \beta_k)$-*weakly DR-modular* if it satisfies both properties.

The parameters $\alpha_k, \beta_k$ characterize how close a set function is to being submodular and supermodular, respectively. If $F$ is non-decreasing, then $\alpha_k, \beta_k \in [0, 1]$, $F$ is submodular (supermodular) if and only if $\alpha_k = 1$ ($\beta_k = 1$) for all $k \in \mathbb{N}_+$, and modular if and only if both $\alpha_k = \beta_k = 1$ for all $k \in \mathbb{N}_+$. The notion of weak DR-submodularity is a stronger notion of approximate submodularity than weak submodularity, as $\gamma_{S,k} \ge \alpha_{|S|+k-1}$ for all $S \subseteq V, k \in N_+$ [El Halabi et al., 2018, Prop. 8]. This implies that GREEDY achieves a $(1 - e^{-\alpha_{2k-1}})$-approximation when $F$ is $\alpha_{2k-1}$-weakly DR-submodular.

A stronger approximation guarantee can be obtained with the notion of *total curvature* introduced in [Sviridenko et al., 2017], which is a stronger notion of approximate submodularity than weak DR-modularity.

**Definition D.2** (Total curvature). Given a set function $F$, we define its total curvature $c_k$ where $k \in \mathbb{N}_+$, as

$$c_k = 1 - \min_{|S| \le k, |T| \le k, i \in V \setminus T} \frac{F(i|S)}{F(i|T)}.$$

Note that if $F$ has total curvature $c_k$, then $F$ is $(1 - c_k, 1 - c_k)$-weakly DR-modular. Given a non-decreasing function $F$ with total curvature $c_k$, the GREEDY algorithm is guaranteed to return a solution $F(\hat{S}) \geq (1 - c_k) \max_{|S| \leq k} F(S)$ [Sviridenko et al., 2017, Theorem 6].

## D.2 Approximate modularity of reweighted input change

The reweighted input change objective $F$ is closely related to the column subset selection objective (the latter is a special case of $F$ where $W^{\ell+1}$ is the identity matrix), whose total curvature was shown to be related to the condition number of $A^\ell$ in [Sviridenko et al., 2017].

We show in Propositions D.3 and D.4 that the total curvatures of $\tilde{F}$ and $F$ are related to the condition number of $A^\ell W^{\ell+1}$.

**Proposition D.3.** *Given $k \in \mathbb{N}_+$, $\tilde{F}$ is a normalized non-decreasing $\alpha_k$-weakly DR-submodular function, with*

$$\alpha_k \geq \frac{\min_{\|z\|_2=1} \|(A^\ell W^{\ell+1})^\top z\|_2^2}{\max_{\|z\|_2=1} \|(A^\ell W^{\ell+1})^\top z\|_2^2}.$$

*Moreover, if any collection of $k + 1$ columns of $B^\ell$ are linearly independent, $\tilde{F}$ is also $\alpha_k$-weakly DR-supermodular and has total curvature $1 - \alpha_k$.*

*Proof.* We adapt the proof from [Sviridenko et al., 2017, Lemma 6]. For all $S \subseteq V, i \in V \setminus S$, we have $F(i|S) = \sum_{m=1}^{n_{\ell+1}} \|\text{proj}_{R_S(b_i^\ell)}(A^\ell) w_m^{\ell+1}\|_2^2$ by Lemma B.5. For all $j \notin S$, we have $\text{proj}_{R_S(b_i^\ell)}(a_j^\ell) = R_S(b_i^\ell) \frac{R_S(b_i^\ell)^\top a_j^\ell}{\|R_S(b_i^\ell)\|^2}$ if $\|R_S(b_i^\ell)\| > 0$, and 0 otherwise, by optimality conditions. Hence, we can write for all $i$ such that $\|R_S(b_i^\ell)\| > 0$,

$$\tilde{F}(i|S) = \sum_{m=1}^{n_{\ell+1}} \|\text{proj}_{R_S(b_i)}(A^\ell) w_m^{\ell+1}\|_2^2 = \sum_{m=1}^{n_{\ell+1}} \| \sum_{j \in V} w_{jm}^{\ell+1} R_S(b_i^\ell) \frac{R_S(b_i^\ell)^\top a_j^\ell}{\|R_S(b_i^\ell)\|^2}\|_2^2 = \|(A^\ell W^{\ell+1})^\top \frac{R_S(b_i^\ell)}{\|R_S(b_i^\ell)\|}\|_2^2$$

Hence,

$$\min_{\|z\|_2=1} \|(A^\ell W^{\ell+1})^\top z\|_2^2 \leq \tilde{F}(i|S) \leq \max_{\|z\|_2=1} \|(A^\ell W^{\ell+1})^\top z\|_2^2$$

Let $v_j = x^S(B^\ell)_{ij}$ for $j \in S$, $v_i = -1$, and $z = v/\|v\|_2$, then $\|v\|_2 \geq 1$, and

$$\|R_S(b_i^\ell)\|^2 = \|B^\ell v\|_2^2 \geq \|B^\ell z\|_2^2 \geq \min_{\|z\|_2 \leq 1, \|z\|_0 \leq |S|+1} \|B^\ell z\|_2^2$$

The bound on $\alpha_k$ then follows by noting that $\|R_S(b_i^\ell)\| \geq \|R_T(b_i^\ell)\|$ for all $S \subseteq T$. The rest of the proposition follows by noting that if any collection of $k + 1$ columns of $B^\ell$ are linearly independent, then $\|R_S(b_i^\ell)\| \geq \min_{\|z\|_2 \leq 1, \|z\|_0 \leq k+1} \|B^\ell z\|_2^2 > 0$ for any $S$ such that $|S| \leq k$. $\square$

As discussed in Section D.1, Proposition D.3 implies that GREEDY achieves a $(1 - e^{-\alpha_{2k-1}})$-approximation with $\tilde{F}(S)$, where $\alpha_k$ is non-zero if all rows of $A^\ell W^{\ell+1}$ are linearly independent. If in addition any $k + 1$ columns of $B^\ell$ are linearly independent, then GREEDY achieves an $\alpha_k$-approximation.

**Proposition D.4.** *Given $k \in \mathbb{N}_+$, $F$ is a normalized non-decreasing $\alpha_k$-weakly DR-submodular function, with*

$$\alpha_k \geq \frac{\max\{\min_{\|z\|_2=1} \|(A^\ell W^{\ell+1})^\top z\|_2^2, \min_{\|z\|_2=1} \|(W^{\ell+1})^\top z\|_2^2 \min_{\|z\|_2 \leq 1, \|z\|_0 \leq k+1} \|A^\ell z\|_2^2\}}{\max_{\|z\|_2=1} \|(A^\ell W^{\ell+1})^\top z\|_2^2}.$$

*Moreover, if any collection of $k + 1$ columns of $A^\ell$ are linearly independent, $F$ is also $\alpha_k$-weakly DR-supermodular and has total curvature $1 - \alpha_k$.*

*Proof.* Setting $B^\ell = A^\ell$ in Lemma B.5, we get

$$\alpha_k \geq \frac{\min_{\|z\|_2=1} \|(A^\ell W^{\ell+1})^\top z\|_2^2}{\max_{\|z\|_2=1} \|(A^\ell W^{\ell+1})^\top z\|_2^2}$$

To obtain the second lower bound, we note that by Lemma B.6 we can write for all $S \subseteq V, i \in V \setminus S$ such that $\|R_S(a_i^\ell)\| > 0$,

$$F(i|S) = \sum_{m=1}^{n_{\ell+1}} (\sum_{j \notin S} w_{jm}^{\ell+1} \frac{R_S(a_i^\ell)^\top a_j^\ell}{\|R_S(a_i^\ell)\|^2})^2 \|R_S(a_i^\ell)\|_2^2 = \|(W^{\ell+1})^\top (A_{V \setminus S}^\ell)^\top \frac{R_S(a_i^\ell)}{\|R_S(a_i^\ell)\|}\|_2^2 \|R_S(a_i^\ell)\|_2^2.$$

Note that $\gamma^{S,i}(a_i) = \frac{R_S(a_i^\ell)^\top a_i^\ell}{\|R_S(a_i^\ell)\|^2} = 1$ by optimality conditions $(R_S(a_i^\ell)^\top a_i^\ell = R_S(a_i^\ell)^\top (R_S(a_i^\ell) + A_S x^S(ai)) = \|R_S(a_i^\ell)\|_2^2$; see proof of Lemma B.4), hence $\|(A_{V \setminus S}^\ell)^\top \frac{R_S(a_i^\ell)}{\|R_S(a_i^\ell)\|}\|_2^2 \geq 1$ and $\|(W^{\ell+1})^\top (A_{V \setminus S}^\ell)^\top \frac{R_S(a_i^\ell)}{\|R_S(a_i^\ell)\|}\|_2^2 \geq \min_{\|z\|_2=1, \|z\|_0 \leq |V \setminus S|} \|(W^{\ell+1})^\top z\|_2^2$. Let $v_j = x^S(A^\ell)_{ij}$ for $j \in S$, $v_i = -1$, and $z = v/\|v\|_2$, then $\|v\|_2 \geq 1$, and

$$\|R_S(a_i^\ell)\|^2 = \|A^\ell v\|_2^2 \geq \|A^\ell z\|_2^2 \geq \min_{\|z\|_2 \leq 1, \|z\|_0 \leq |S|+1} \|A^\ell z\|_2^2$$

We thus have $F(i|S) \geq \min_{\|z\|_2=1, \|z\|_0 \leq |V \setminus S|} \|(W^{\ell+1})^\top z\|_2^2 \min_{\|z\|_2 \leq 1, \|z\|_0 \leq |S|+1} \|A^\ell z\|_2^2$.

The bound on $\alpha_k$ then follows by noting that $\|R_S(a_i^\ell)\| \geq \|R_T(a_i^\ell)\|$ for all $S \subseteq T$. The rest of the proposition follows by noting that if any collection of $k+1$ columns of $A^\ell$ are linearly independent, then $\|R_S(a_i^\ell)\| \geq \min_{\|z\|_2 \leq 1, \|z\|_0 \leq k+1} \|A^\ell z\|_2^2 > 0$ for any $S$ such that $|S| \leq k$. $\qquad \square$

As discussed in Section D.1, Proposition D.3 implies that GREEDY achieves a $(1 - e^{-\alpha_{2k-1}})$-approximation with $F(S)$, where $\alpha_k$ is non-zero if all rows of $A^\ell W^{\ell+1}$ are linearly independent. Moreover, if any $k+1$ columns of $A^\ell$ are linearly independent and all rows of $W^{\ell+1}$ are linearly independent, then GREEDY achieves an $\alpha_k$-approximation.

It is worth noting that if $W^{\ell+1}$ is identity matrix, i.e., $F$ is the column subset selection function, then Proposition D.4 implies a stronger result than [Sviridenko et al., 2017, Lemma 6] for some cases. In particular, we have

$$\alpha_k \geq \frac{\max\{\min_{\|z\|_2 \leq 1, \|z\|_0 \leq k+1} \|A^\ell z\|_2^2, \min_{\|z\|_2=1} \|(A^\ell)^\top z\|_2^2\}}{\max_{\|z\|_2 \leq 1} \|(A^\ell)^\top z\|_2^2\}},$$

which implies that $F$ is weakly DR-submodular if any $k$ columns of $A^\ell$ are linearly independent, or if all rows of $A^\ell$ are linearly independent.

## E  Empirical values of the submodularity ratio

As discussed in Section 3.1, computing the lower bounds on the submodularity ratio $\gamma_{U,k}$ in Proposition 3.1 and 4.1 is NP-Hard [Das and Kempe, 2011] ($\min_{\|z\|_2=1, \|z\|_0 \leq |U|+k} \|A^\ell z\|_2^2$ corresponds to $\lambda_{\min}(C, |U|+k)$ in their notation). One simple lower bound that can be obtained from the eigenvalue interlacing theorem is $\gamma_{U,k} \geq \frac{\lambda_{\min}((A^\ell)^\top A^\ell)}{\lambda_{\max}((A^\ell)^\top A^\ell)}$. However, such bound is too loose as it is often equal to zero. For this reason, we focus on when our lower bounds on $\gamma_{\hat{S},k}$ are non-zero: the lower bounds in Proposition 3.1 and 4.1 are non-zero if any $\min\{2k, n_\ell\}$ and $\min\{2k, n_\ell\} r_h r_w$ columns of $A^\ell$ are linearly independent, respectively. We report in Table 1 an upper bound on $k$ for which these conditions hold, based on the rank of the activation matrix $A^\ell$, in each pruned layer in the three models we used in our experiments.

We observe that the upper bound is close to $0.5$ for most linear layers, but is very small in some convolution layers (e.g., features.15, 18, 22 in VGG11). As explained in Section 4, the linear independence condition required for convolution layers (Proposition 4.1) only holds for very small $k$, due to the correlation between patches which overlap. This can be avoided in most layers by sampling $r_h r_w$ random patches from each image (to ensure $A^\ell$ is a tall matrix), instead of using all patches. We report in Table 2 the corresponding upper bounds on $k$ in this setting for the VGG11 model.

The upper bounds are indeed larger than the ones obtained with all patches. However, some layers (e.g., features.15, 18, 22) have a very small feature map size (respectively $4 \times 4, 4 \times 4, 2 \times 2$) so that even the small number of random patches have significant overlap, resulting in still a very small upper bound. Our experiments with random patches on VGG11 yielded worst results, so we chose

Table 1: Largest possible $k$ for which our lower bounds on the submodularity ratio $\gamma_{\hat{S},k}$ are non-zero, when using all patches per image.

| Dataset | Model | upper bound on $k/n_\ell$ (all patches, $n = 512$) |
|---------|-------|-----------------------------------------------------|
| MNIST | LeNet | conv1: 0.37, conv2: 1, fc1: 0.46, fc2: 0.49 |
| CIFAR10 | VGG11 | features.0: 0.22, features.4: 0.39, features.8: 0.25, features.11: 0.28, features.15: 0.06, features.18: 0.03, features.22: 0.01, classifier.0: 1, classifier.3: 1 |
| CIFAR10 | ResNet56 | layer1.0.conv1: 0.12, layer1.1.conv1: 0.15, layer1.2.conv1: 0.22, layer1.3.conv1: 0.22, layer1.4.conv1: 0.30, layer1.5.conv1: 0.28, layer1.6.conv1: 0.44, layer1.7.conv1: 0.35, layer1.8.conv1: 0.15, layer2.0.conv1: 0.48, layer2.1.conv1: 0.47, layer2.2.conv1: 0.48, layer2.3.conv1: 0.47, layer2.4.conv1: 0.48, layer2.5.conv1: 0.47, layer2.6.conv1: 0.48, layer2.7.conv1: 0.47, layer2.8.conv1: 0.48, layer3.0.conv1: 0.47, layer3.1.conv1: 0.49, layer3.2.conv1: 0.46, layer3.3.conv1: 0.48, layer3.4.conv1: 0.48, layer3.5.conv1: 0.49, layer3.6.conv1: 0.48, layer3.7.conv1: 0.48, layer3.8.conv1: 1 |

Table 2: Largest possible $k$ for which our lower bounds on the submodularity ratio $\gamma_{\hat{S},k}$ are non-zero, in VGG11 on CIFAR10, when using $r_h r_w$ random patches per image.

| Dataset | Model | upper bound on $k/n_\ell$ (random patches, $n = 512$) |
|---------|-------|-------------------------------------------------------|
| CIFAR10 | VGG11 | features.0: 0.38, features.4: 0.43, features.8: 0.29, features.11: 0.31, features.15: 0.15, features.18: 0.08, features.22: 0.1, classifier.0: 1, classifier.3: 1 |

to use all patches. Note that our lower bounds on $\gamma_{\hat{S},k}$ are not necessarily tight (see Appendix F). Hence, if $k$ is outside these ranges, our lower bound on $\gamma_{\hat{S},k}$ is zero, but not necessarily $\gamma_{\hat{S},k}$ itself; indeed in our experiments our methods still perform well in these cases.

## F  Tightness of lower bounds on the submodularity ratio

In this section, we investigate how tight are the lower bounds on the submodularity ratio $\gamma_{U,k}$ in Propositions 3.1 and 4.1. One trivial example where these bounds are tight is when $A^\ell$ is the identity matrix. In this case, both $F$ and $G$ are submodular, hence their corresponding $\gamma_{U,k} = 1$ for all $U$ and $k$, and the lower bounds in both Proposition 3.1 and 4.1 are also equal to one. We present below another more interesting example where the bounds are tight.

**Proposition F.1.** *Given any matrix $A^\ell$ whose columns have equal norm, there exists a matrix $W^{\ell+1}$ such that the corresponding function $F$ has $\gamma_{\emptyset,k} = \frac{\min_{\|z\|_2=1,\|z\|_0\leq k} \|A^\ell z\|_2^2}{\max_{\|z\|_2=1,\|z\|_0\leq 1} \|A^\ell z\|_2^2}$.*

*Proof.* Given a set $S$, let $v_{\min}^S, u_{\min}^S$ be the right and left singular vectors of $A_S^\ell$ corresponding to the smallest singular value $\sigma_{\min}^S$ of $A_S^\ell$, i.e., $A_S^\ell v_{\min}^S = \sigma_{\min}^S u_{\min}^S$ and $(A_S^\ell)^\top u_{\min}^S = \sigma_{\min}^S v_{\min}^S$. We consider the case where all columns of $A^\ell$ have equal norm, which we denote by $\sigma_{\max}^1$, and $W^{\ell+1}$ have all columns equal to $c v_{\min}^S$ for some scalar $c > 0$. Then for all $m$, $A_S^\ell w_m^{\ell+1} = c\sigma_{\min}^S u_{\min}^S$ and the minimum of $\min_{\text{supp}(x)\subseteq S} \|c\sigma_{\min}^S u_{\min}^S - A^\ell x\|_2^2$ is obtained at $x_S = c v_{\min}^S$. We can thus write

$$F(S) = \sum_{m=1}^{n_{\ell+1}} \|c\sigma_{\min}^S u_{\min}^S\|_2^2 - \min_{\text{supp}(x)\subseteq S} \|c\sigma_{\min}^S u_{\min}^S - A^\ell x\|_2^2$$

$$= \sum_{m=1}^{n_{\ell+1}} (c\sigma_{\min}^S)^2 - \|c\sigma_{\min}^S u_{\min}^S - c\sigma_{\min} u_{\min}^S\|_2^2$$

$$= n_{\ell+1} (c\sigma_{\min}^S)^2$$

On the other hand, for any $i \in S$, let $y = c\sigma_{\min}^S u_{\min}^S$, the minimum of $\min_{\text{supp}(x) \subseteq \{i\}} \|y - A^\ell x\|_2^2$ is obtained at $x_i = \frac{(a_i^\ell)^\top y}{\|a_i^\ell\|_2^2}$. We have

$$
\begin{aligned}
F(i) &= \sum_{m=1}^{n_{\ell+1}} \|y\|_2^2 - \min_{\text{supp}(x) \subseteq \{i\}} \|y - A^\ell x\|_2^2 \\
&= \sum_{m=1}^{n_{\ell+1}} \|y\|_2^2 - \|y - a_i^\ell \frac{(a_i^\ell)^\top y}{\|a_i^\ell\|_2^2}\|_2^2 \\
&= n_{\ell+1} \frac{((a_i^\ell)^\top y)^2}{\|a_i^\ell\|_2^2}
\end{aligned}
$$

Hence,

$$
\begin{aligned}
\sum_{i \in S} F(i) &= n_{\ell+1} \sum_{i \in S} \frac{((a_i^\ell)^\top y)^2}{\|a_i^\ell\|_2^2} \\
&= n_{\ell+1} \frac{\|(A_S^\ell)^\top y\|_2^2}{(\sigma_{\max}^1)^2} \\
&= n_{\ell+1} \frac{c^2 (\sigma_{\min}^S)^4}{(\sigma_{\max}^1)^2}
\end{aligned}
$$

Then

$$
\gamma_{\emptyset, k} \leq \frac{\sum_{i \in S} F(i)}{F(S)} = \frac{(\sigma_{\min}^S)^2}{(\sigma_{\max}^1)^2} = \frac{\min_{\|z\|_2=1, \text{supp}(z)=S} \|A^\ell z\|_2^2}{\max_{\|z\|_2=1, \|z\|_0=1} \|A^\ell z\|_2^2}.
$$

If we choose $S$ such that $S \in \arg\min_{|S| \leq k} (\sigma_{\min}^S)^2$, we get $\gamma_{\emptyset, k} = \frac{\min_{\|z\|_2=1, \|z\|_0 \leq k} \|A^\ell z\|_2^2}{\max_{\|z\|_2=1, \|z\|_0 \leq 1} \|A^\ell z\|_2^2}$ by Proposition 3.1. $\qquad \square$

Since $F$ is a special case of $G$ where $M$ is the identity map, the above example applies to $G$ too. On the other hand, there are also cases where these bounds are not tight. In particular, there are cases where the lower bound on $\alpha_{|U|+k-1}$ in Proposition D.4 is larger than the lower bound on $\gamma_{U,k}$ in Proposition 3.1, which implies that the latter is not tight since $\gamma_{U,k} \geq \alpha_{|U|+k-1}$ (see Section D.1). For example, if all rows of $A^\ell W^{\ell+1}$ are linearly independent, but there exists $2k$ columns of $A^\ell$ which are linearly dependent, then the bound in Proposition 3.1 is zero while the one in Proposition D.4 is not. These borderline cases are unlikely to occur in practice. Whether we can tighten these lower bounds based on realistic assumptions on the weights and activations is an interesting future research question.

## G    Experimental setup

Our code uses Pytorch [Paszke et al., 2017] and builds on the open source ShrinkBench library introduced in [Blalock et al., 2020]. We use the code from [Buschjäger et al., 2020] for GREEDY. Our implementation of LAYERSAMPLING is adapted from the code provided in [Liebenwein et al., 2020]. We implemented the version of LAYERGREEDYFS implemented in the code of [Ye et al., 2020a], which differs from the version described in the paper: added neurons/channels are not allowed to be repeated. We use the implementation of LeNet and ResNet56 included in ShrinkBench [Blalock et al., 2020], and a modified version of the implementation of VGG11 provided in [Phan, 2021], where we changed the number of neurons in the first two layers to 128.

We conducted experiments on 3 different clusters with the following resources (per experiment):

- Cluster 1: $1 \times$ NVidia A100 with 40G memory, $20 \times$ AMD Milan 7413 @ 2.65 GHz / AMD Rome 7532 @ 2.40 GHz
- Cluster 2: $1 \times$ NVIDIA P100 Pascal with 12G/16G memory / NVIDIA V100 Volta with 32G memory, $20 \times$ Intel CPU of various types
- Cluster 3: $1 \times$ NVIDIA Quadro RTX 8000 with 48G memory / NVIDIA Tesla M40 with 24G memory / NVIDIA TITAN RTX with 24G memory, $6 \times$ CPU of various types.

Pruning and fine-tuning with limited data was done on CPUs for all methods, a GPU was used only when fine-tuning with full data.

All our experiments used the following setup:

Random seeds: $42, 43, 44, 45, 46$

Pruning setup:

- Number of Batches: 4 (sampled at random from the training set)
- Batch size: 128
- Values used for compression ratio:
$$c \in \{1, 2, 4, 8, 16, 32, 64, 128\}$$
- Values used for per-layer fraction selection:
$$\alpha_\ell \in \{0.01, 0.05, 0.075, 0.1, 0.15, 0.2, \cdots, 0.95, 1.0\}$$
- Verification set used for the budget selection method in Section 5.2: random subset of training set of same size as validation set.

Training and fine-tuning setup for LeNet on MNIST:

- Batch size: 128
- Epochs for pre-training: 200
- Epochs for fine-tuning: 10
- Optimizer for pre-training: SGD with Nestrov momentum 0.9
- Optimizer for fine-tuning: Adam with $\beta_1 = 0.9$ and $\beta_2 = 0.99$
- Initial learning rate: $1 \times 10^{-3}$
- Learning rate schedule: Fixed

Training and fine-tuning setup for VGG11 on CIFAR10:

- Batch size: 128
- Epochs for pre-training: 200
- Epochs for fine-tuning: 20
- Optimizer for pre-training: Adam with $\beta_1 = 0.9$ and $\beta_2 = 0.99$
- Optimizer for fine-tuning: Adam with $\beta_1 = 0.9$ and $\beta_2 = 0.99$
- Initial learning rate: $1 \times 10^{-3}$
- Weight decay: $5 \times 10^{-4}$
- Learning rate schedule for pre-training: learning rate dropped by 0.1 at epochs 100 and 150
- Learning rate schedule for fine-tuning: learning rate dropped by 0.1 at epochs 10 and 15

The setup for fine-tuning ResNet56 on CIFAR10 is the same one used for VGG, as outlined above.

## H  Effect of fine-tuning

In this section, we study the effect of fine-tuning with both limited and sufficient data. To that end, we report the top-1 accuracy results of all the pruning tasks considered in Section 7, after fine-tuning with only four batches of training data, and after fine-tuning with the full training data in Figure 3. We fine-tune for 10 epochs in the MNIST experiment, and for 20 epochs in both CIFAR-10 experiments. We do not fine-tune at compression ratio 1 (i.e., when nothing is pruned).

Our method still outperforms other baselines after fine-tuning with limited-data, and is among the best performing methods even in the full-data setting (if we consider the non-reweighted variants for VGG11 model). As expected, fine-tuning with the full training data provides a significant boost

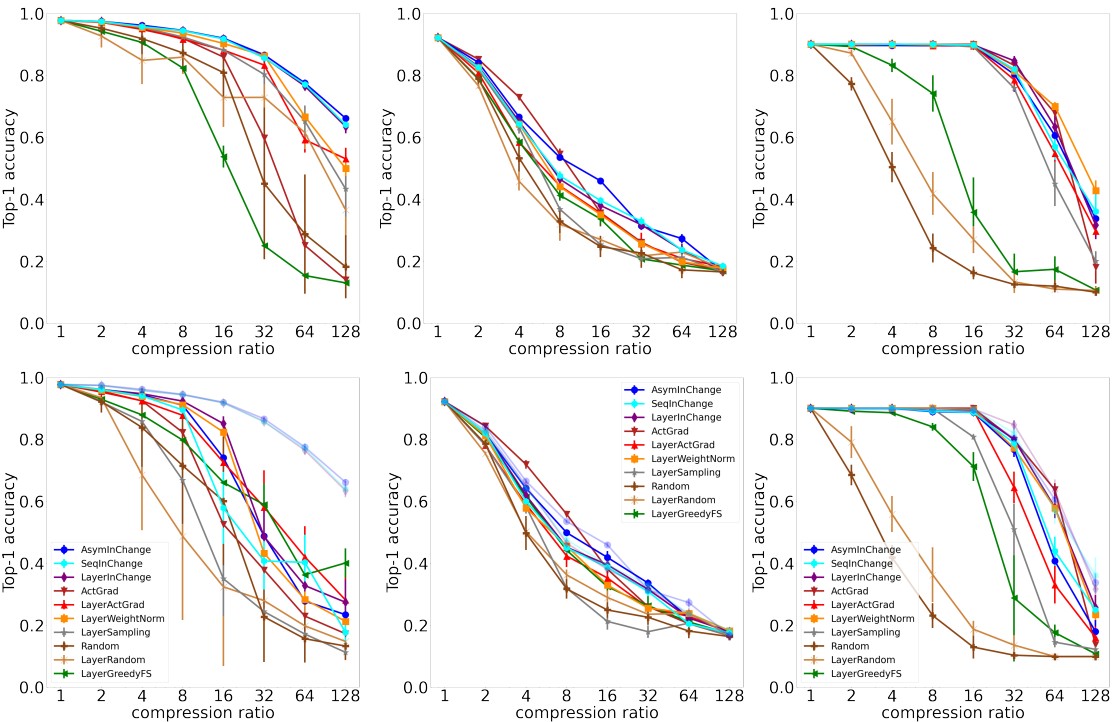

Figure 2: Top-1 Accuracy of different pruning methods, after fine-tuning with four batches of training data, applied to LeNet on MNIST (left), ResNet56 on CIFAR10 (middle), and VGG11 on CIFAR10 (right), for several compression ratios (in log-scale), with (top) and without (bottom) reweighting. We include the three reweighted variants of our method in the bottom plots (faded) for reference.

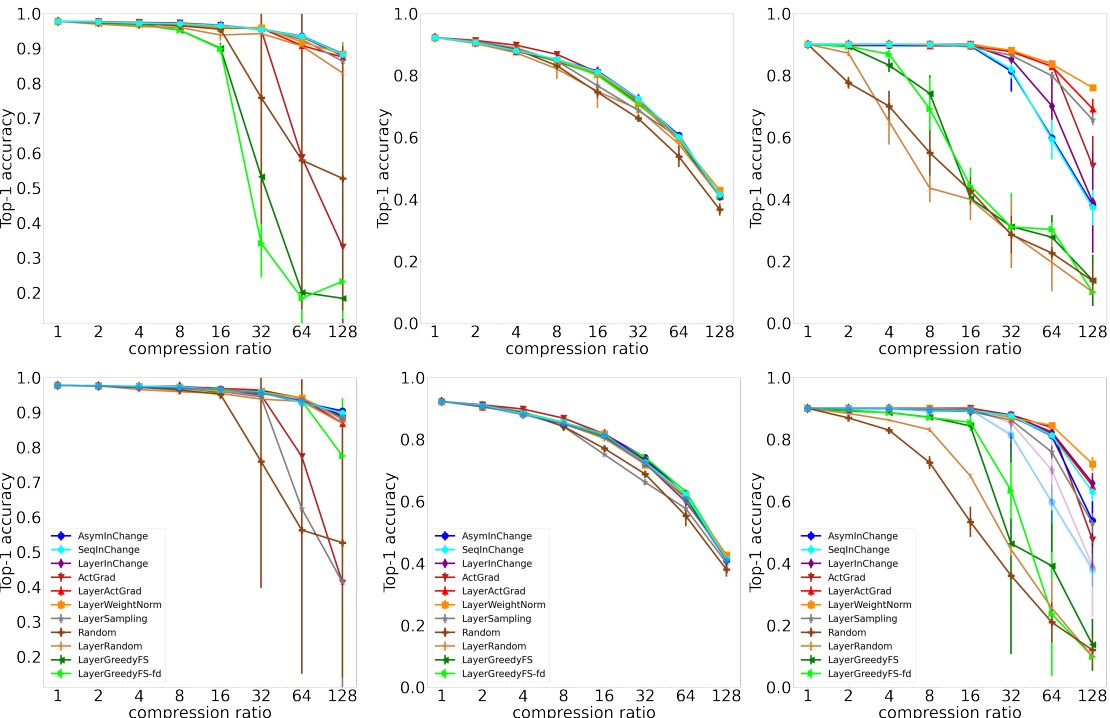

Figure 3: Top-1 Accuracy of different pruning methods, after fine-tuning with the full training data, applied to LeNet on MNIST (left), ResNet56 on CIFAR10 (middle), and VGG11 on CIFAR10 (right), for several compression ratios (in log-scale), with (top) and without (bottom) reweighting. We include the three reweighted variants of our method in the bottom plots (faded) for reference.

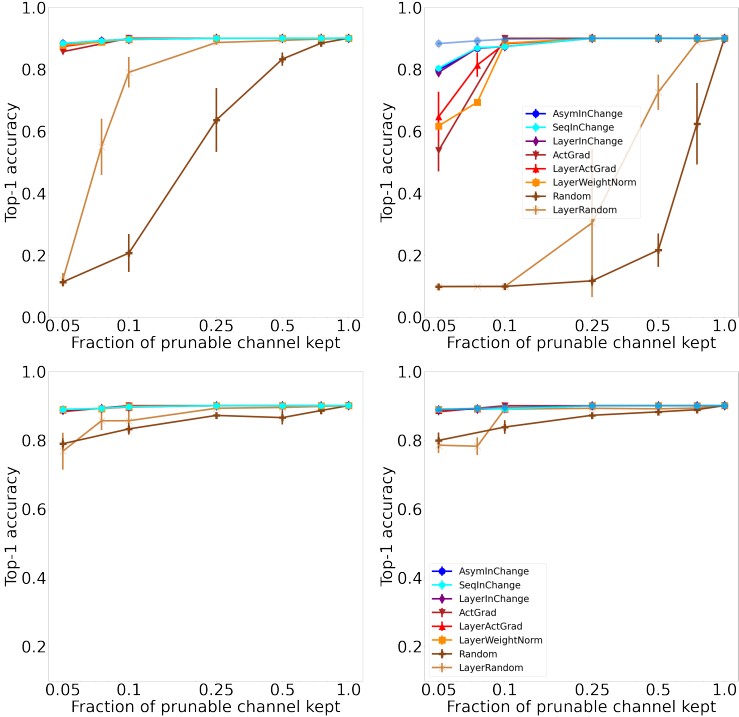

Figure 4: Top-1 Accuracy of different pruning methods on CIFAR10, after pruning the first and second to last convolution layers in VGG11 model, with different fractions of remaining channels (in log-scale), with (left) and without (right) reweighting, with (bottom) and without (top) fine-tuning, with per-layer fractions selected using the selection method discussed in Section 5.2. We include the three reweighted variants of our method in the plots without reweighting (faded) for reference.

in performance to all methods, even more than reweighting. Fine-tuning with limited data also helps but significantly less. Reweighting still improves the performance of all methods, except LAYERGREEDYFS and LAYERGREEDYFS-fd, even when fine-tuning with limited-data is used, but it can actually deteriorate performance when fine-tuning with full-data is used (see Figure 3 left-bottom plot, the reweighted variants of our methods have lower accuracy than the non-reweighted variants). We suspect that this could be due to overfitting to the very small training data used for pruning. This only happens with VGG11 model, because it is larger than LeNet and ResNet56 models. We expect the performance of the reweighted variants of our method to improve if we use more training data for pruning.

# I  Importance of per-layer budget selection

In this section, we study the effect of per-layer budget selection on accuracy. To that end, we use the same pretrained VGG11 model from Section 7, and prune the first and second to last convolution layers in it. Since the second to last layer in VGG11 (features.22) has little effect on accuracy when pruned, we expect the choice of how the global budget is distributed on the two layers to have a significant impact on performance. Figure 4 shows the top-1 accuracy for different fractions of prunable channels kept, when the per-layer budget selection from Section 5.2 is used, while Figure 5 shows the results when equal fractions of channels kept are used in each layer. As expected, all layerwise methods perform much more poorly with equal per-layer fractions, both with and without fine-tuning. Though, the difference is less drastic when fine-tuning is used.

# J  Results with respect to other metrics

We report in Tables 3, 4 and 5, the top-1 accuracy, speedup ratio ($\frac{\text{original number of FLOPs}}{\text{pruned number of FLOPs}}$), and pruning time values of the experiments presented in Section 7. We exclude the worst performing methods RANDOM and LAYERRANDOM.

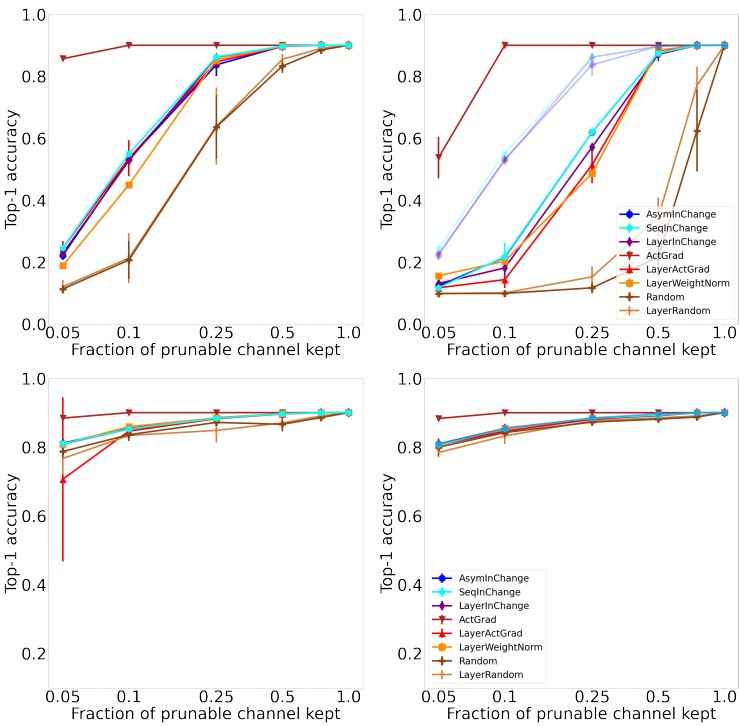

Figure 5: Top-1 Accuracy of different pruning methods on CIFAR10, after pruning the first and second to last convolution layers in VGG11 model, with different fractions of remaining channels (in log-scale), with (left) and without (right) reweighting, with (bottom) and without (top) fine-tuning, with equal per-layer fractions. We include the three reweighted variants of our method in the plots without reweighting (faded) for reference.

Note that for a given compression ratio speedup values vary significantly between different pruning methods, because the number of weights and flops vary between layers, and pruning methods differ in their per-layer budget allocations. The best performing methods in terms of compression are not necessarily the best ones in terms of speedup. For example, ACTGRAD is among the best performing methods in terms of compression on ResNet56-CIFAR10, but it is the worst one in terms of speedup. In cases where we care more about speedup than compression, we can replace the constraint in the per-layer budget selection problem (6) to be speedup instead of compression.

Since our goal in these experiments was to study performance in terms of accuracy vs compression rate, we did not focus on optimizing our method's implementation for computation time efficiency. For example, our current implementation uses the classical Greedy algorithm. This can be significantly sped-up by switching to the faster Greedy algorithm from Li et al. [2022].

Table 3: Top-1 Accuracy % (Acc1), speedup ratio (SR), and pruning time (in hrs:mins:secs) of different pruning methods applied to LeNet on MNIST, with different compression ratios $c$, with and without reweighting (rw) and fine-tuning (ft).

| Method | rw | ft | c=2 Acc1 | SR | time | c=4 Acc1 | SR | time | c=8 Acc1 | SR | time | c=16 Acc1 | SR | time | c=32 Acc1 | SR | time |
|---|---|---|---|---|---|---|---|---|---|---|---|---|---|---|---|---|---|
| AsymInChange | ✓ | ✓ | 97.5 | 3.1 | 0:00:03 | 97.1 | 7.3 | 0:00:02 | 97.0 | 6.2 | 0:00:02 | 96.6 | 8.3 | 0:00:02 | 95.4 | 4.0 | 0:00:02 |
| | ✓ | ✗ | 97.4 | 3.1 | 0:00:03 | 96.2 | 7.3 | 0:00:02 | 94.4 | 6.2 | 0:00:02 | 90.3 | 8.3 | 0:00:02 | 83.5 | 4.0 | 0:00:02 |
| | ✗ | ✓ | 97.6 | 3.1 | 0:00:03 | 97.4 | 4.0 | 0:00:02 | 97.4 | 2.4 | 0:00:02 | 96.2 | 8.3 | 0:00:01 | 95.7 | 3.5 | 0:00:02 |
| | ✗ | ✗ | 84.6 | 3.1 | 0:00:03 | 48.7 | 4.0 | 0:00:02 | 36.4 | 2.4 | 0:00:02 | 12.8 | 8.3 | 0:00:01 | 14.4 | 3.5 | 0:00:02 |
| SeqInChange | ✓ | ✓ | 97.6 | 3.1 | 0:00:03 | 97.2 | 7.3 | 0:00:02 | 97.1 | 6.2 | 0:00:02 | 96.5 | 8.3 | 0:00:02 | 95.4 | 4.0 | 0:00:02 |
| | ✓ | ✗ | 97.4 | 3.1 | 0:00:03 | 95.8 | 7.3 | 0:00:02 | 94.3 | 6.2 | 0:00:02 | 89.7 | 8.3 | 0:00:02 | 82.2 | 4.0 | 0:00:02 |
| | ✗ | ✓ | 97.6 | 3.1 | 0:00:04 | 97.5 | 4.0 | 0:00:02 | 97.3 | 2.4 | 0:00:03 | 96.5 | 8.3 | 0:00:01 | 95.8 | 3.5 | 0:00:01 |
| | ✗ | ✗ | 88.5 | 3.1 | 0:00:04 | 39.0 | 4.0 | 0:00:02 | 25.8 | 2.4 | 0:00:03 | 12.4 | 8.3 | 0:00:02 | 10.7 | 3.5 | 0:00:01 |
| LayerInChange | ✓ | ✓ | 97.5 | 3.1 | 0:00:03 | 97.1 | 7.3 | 0:00:02 | 97.0 | 6.2 | 0:00:02 | 96.6 | 8.3 | 0:00:02 | 95.6 | 4.0 | 0:00:01 |
| | ✓ | ✗ | 97.4 | 3.1 | 0:00:03 | 95.8 | 7.3 | 0:00:02 | 94.2 | 6.2 | 0:00:02 | 89.9 | 8.3 | 0:00:02 | 81.8 | 4.0 | 0:00:01 |
| | ✗ | ✓ | 97.6 | 3.1 | 0:00:03 | 97.4 | 4.0 | 0:00:02 | 97.5 | 2.4 | 0:00:02 | 96.5 | 8.3 | 0:00:02 | 95.6 | 3.5 | 0:00:01 |
| | ✗ | ✗ | 88.6 | 3.1 | 0:00:03 | 40.6 | 4.0 | 0:00:02 | 33.5 | 2.4 | 0:00:02 | 23.9 | 8.3 | 0:00:02 | 13.1 | 3.5 | 0:00:01 |
| ActGrad | ✓ | ✓ | 97.7 | 2.0 | 0:00:00 | 97.4 | 2.6 | 0:00:00 | 97.3 | 3.4 | 0:00:00 | 96.6 | 4.1 | 0:00:00 | 95.3 | 4.8 | 0:00:00 |
| | ✓ | ✗ | 97.2 | 2.0 | 0:00:00 | 94.7 | 2.6 | 0:00:00 | 87.2 | 3.4 | 0:00:00 | 67.5 | 4.1 | 0:00:00 | 40.5 | 4.8 | 0:00:00 |
| | ✗ | ✓ | 97.6 | 2.0 | 0:00:00 | 97.5 | 2.6 | 0:00:00 | 97.2 | 3.4 | 0:00:00 | 96.6 | 4.1 | 0:00:00 | 95.0 | 4.8 | 0:00:00 |
| | ✗ | ✗ | 81.0 | 2.0 | 0:00:00 | 43.5 | 2.6 | 0:00:00 | 24.9 | 3.4 | 0:00:00 | 17.7 | 4.1 | 0:00:00 | 15.1 | 4.8 | 0:00:00 |
| LayerActGrad | ✓ | ✓ | 97.6 | 3.1 | 0:00:00 | 97.1 | 4.7 | 0:00:00 | 96.6 | 11.2 | 0:00:00 | 95.8 | 16.2 | 0:00:00 | 95.8 | 4.0 | 0:00:00 |
| | ✓ | ✗ | 97.1 | 3.1 | 0:00:00 | 95.1 | 4.7 | 0:00:00 | 90.2 | 11.2 | 0:00:00 | 82.9 | 16.2 | 0:00:00 | 76.9 | 4.0 | 0:00:00 |
| | ✗ | ✓ | 97.5 | 3.2 | 0:00:00 | 97.4 | 4.4 | 0:00:00 | 97.4 | 2.4 | 0:00:00 | 96.9 | 3.0 | 0:00:00 | 96.4 | 3.1 | 0:00:00 |
| | ✗ | ✗ | 84.9 | 3.2 | 0:00:00 | 46.3 | 4.4 | 0:00:00 | 27.9 | 2.4 | 0:00:00 | 20.5 | 3.0 | 0:00:00 | 17.0 | 3.1 | 0:00:00 |
| LayerWeightNorm | ✓ | ✓ | 97.5 | 3.1 | 0:00:00 | 97.1 | 6.7 | 0:00:00 | 96.7 | 11.2 | 0:00:00 | 95.9 | 16.2 | 0:00:00 | 96.0 | 4.0 | 0:00:00 |
| | ✓ | ✗ | 97.3 | 3.1 | 0:00:00 | 95.5 | 6.7 | 0:00:00 | 93.6 | 11.2 | 0:00:00 | 88.2 | 16.2 | 0:00:00 | 81.1 | 4.0 | 0:00:00 |
| | ✗ | ✓ | 97.5 | 3.0 | 0:00:00 | 97.5 | 3.0 | 0:00:00 | 97.3 | 2.6 | 0:00:00 | 96.0 | 14.7 | 0:00:00 | 95.8 | 3.5 | 0:00:00 |
| | ✗ | ✗ | 91.6 | 3.0 | 0:00:00 | 60.1 | 3.0 | 0:00:00 | 53.9 | 2.6 | 0:00:00 | 18.1 | 14.7 | 0:00:00 | 14.8 | 3.5 | 0:00:00 |
| LayerSampling | ✓ | ✓ | 97.5 | 1.9 | 0:00:20 | 97.4 | 2.7 | 0:00:21 | 97.0 | 3.5 | 0:00:19 | 96.4 | 4.4 | 0:00:20 | 95.7 | 5.2 | 0:00:19 |
| | ✓ | ✗ | 97.2 | 1.9 | 0:00:20 | 95.5 | 2.7 | 0:00:21 | 91.6 | 3.5 | 0:00:19 | 85.5 | 4.4 | 0:00:20 | 71.7 | 5.2 | 0:00:19 |
| | ✗ | ✓ | 97.6 | 1.9 | 0:00:19 | 97.3 | 2.7 | 0:00:20 | 96.8 | 3.5 | 0:00:18 | 96.1 | 4.4 | 0:00:19 | 94.4 | 5.2 | 0:00:18 |
| | ✗ | ✗ | 41.8 | 1.9 | 0:00:19 | 25.5 | 2.7 | 0:00:20 | 22.4 | 3.5 | 0:00:18 | 13.2 | 4.4 | 0:00:19 | 14.1 | 5.2 | 0:00:18 |
| LayerGreedyFS | ✓ | ✓ | 97.3 | 3.3 | 0:00:33 | 96.9 | 4.9 | 0:00:27 | 95.2 | 7.4 | 0:00:40 | 90.0 | 10.4 | 0:00:19 | 53.3 | 9.0 | 0:00:28 |
| | ✓ | ✗ | 94.1 | 3.3 | 0:00:33 | 88.1 | 4.9 | 0:00:27 | 76.2 | 7.4 | 0:00:40 | 44.9 | 10.4 | 0:00:19 | 23.8 | 9.0 | 0:00:28 |
| | ✗ | ✓ | 97.6 | 1.2 | 0:00:27 | 97.4 | 4.2 | 0:00:23 | 97.0 | 2.4 | 0:00:18 | 95.8 | 14.7 | 0:00:14 | 96.2 | 5.1 | 0:00:09 |
| | ✗ | ✗ | 73.5 | 1.2 | 0:00:27 | 45.0 | 4.2 | 0:00:23 | 28.0 | 2.4 | 0:00:18 | 28.3 | 14.7 | 0:00:14 | 20.1 | 5.1 | 0:00:09 |
| LayerGreedyFS-fd | ✓ | ✓ | 97.2 | 3.2 | 0:01:44 | 96.8 | 4.9 | 0:01:21 | 95.2 | 7.4 | 0:01:23 | 89.9 | 10.4 | 0:00:56 | 34.1 | 23.9 | 0:00:48 |
| | ✓ | ✗ | 94.2 | 3.2 | 0:01:44 | 88.2 | 4.9 | 0:01:21 | 75.8 | 7.4 | 0:01:23 | 47.2 | 10.4 | 0:00:56 | 17.2 | 23.9 | 0:00:48 |
| | ✗ | ✓ | 97.6 | 1.2 | 0:01:30 | 97.4 | 4.4 | 0:01:19 | 97.0 | 2.4 | 0:01:01 | 96.7 | 7.5 | 0:00:36 | 95.5 | 3.5 | 0:00:21 |
| | ✗ | ✗ | 73.5 | 1.2 | 0:01:30 | 47.4 | 4.4 | 0:01:19 | 30.3 | 2.4 | 0:01:01 | 23.3 | 7.5 | 0:00:36 | 16.9 | 3.5 | 0:00:21 |

Table 4: Top-1 Accuracy % (Acc1), speedup ratio (SR), and pruning time (in hrs:mins:secs) of different pruning methods applied to ResNet56 on CIFAR10, with different compression ratios $c$, with and without reweighting (rw) and fine-tuning (ft).

| Method | rw | ft | c=2 Acc1 | SR | time | c=4 Acc1 | SR | time | c=8 Acc1 | SR | time | c=16 Acc1 | SR | time | c=32 Acc1 | SR | time |
|---|---|---|---|---|---|---|---|---|---|---|---|---|---|---|---|---|---|
| AsymInChange | ✓ | ✓ | 90.7 | 2.6 | 2:13:10 | 87.9 | 6.0 | 1:14:00 | 84.9 | 11.0 | 0:42:08 | 81.1 | 16.4 | 0:24:43 | 72.1 | 21.3 | 0:14:57 |
| | ✓ | ✗ | 84.2 | 2.6 | 2:13:10 | 46.4 | 6.0 | 1:14:00 | 20.6 | 11.0 | 0:42:08 | 18.3 | 16.4 | 0:24:43 | 10.2 | 21.3 | 0:14:57 |
| | ✗ | ✓ | 90.9 | 2.5 | 2:15:09 | 88.3 | 6.0 | 1:12:51 | 85.4 | 10.4 | 0:40:31 | 81.7 | 15.1 | 0:26:00 | 74.2 | 20.2 | 0:15:23 |
| | ✗ | ✗ | 73.3 | 2.5 | 2:15:09 | 16.4 | 6.0 | 1:12:51 | 13.6 | 10.4 | 0:40:31 | 9.9 | 15.1 | 0:26:00 | 10.7 | 20.2 | 0:15:23 |
| SeqInChange | ✓ | ✓ | 90.7 | 2.6 | 4:13:21 | 88.0 | 6.0 | 2:50:54 | 85.2 | 11.0 | 1:30:23 | 81.1 | 16.4 | 0:48:09 | 72.6 | 21.3 | 0:24:36 |
| | ✓ | ✗ | 82.3 | 2.6 | 4:13:21 | 45.5 | 6.0 | 2:50:54 | 24.6 | 11.0 | 1:30:23 | 17.3 | 16.4 | 0:48:09 | 12.9 | 21.3 | 0:24:36 |
| | ✗ | ✓ | 90.9 | 2.5 | 4:01:33 | 88.3 | 6.0 | 2:41:17 | 85.5 | 10.4 | 1:29:22 | 81.4 | 15.1 | 0:46:34 | 72.7 | 20.2 | 0:22:47 |
| | ✗ | ✗ | 75.5 | 2.5 | 4:01:33 | 17.5 | 6.0 | 2:41:17 | 12.1 | 10.4 | 1:29:22 | 9.9 | 15.1 | 0:46:34 | 10.6 | 20.2 | 0:22:47 |
| LayerInChange | ✓ | ✓ | 90.7 | 2.6 | 4:20:16 | 88.0 | 6.0 | 2:53:00 | 85.1 | 11.0 | 1:38:14 | 81.5 | 16.4 | 0:51:46 | 72.6 | 21.3 | 0:24:11 |
| | ✓ | ✗ | 72.3 | 2.6 | 4:20:16 | 23.7 | 6.0 | 2:53:00 | 15.8 | 11.0 | 1:38:14 | 14.0 | 16.4 | 0:51:46 | 11.3 | 21.3 | 0:24:11 |
| | ✗ | ✓ | 90.9 | 2.5 | 4:21:26 | 88.4 | 6.0 | 2:49:09 | 85.4 | 10.4 | 1:29:47 | 81.9 | 15.1 | 0:49:17 | 73.4 | 20.2 | 0:22:33 |
| | ✗ | ✗ | 38.6 | 2.5 | 4:21:26 | 11.4 | 6.0 | 2:49:09 | 9.9 | 10.4 | 1:29:47 | 10.2 | 15.1 | 0:49:17 | 11.0 | 20.2 | 0:22:33 |
| ActGrad | ✓ | ✓ | 91.3 | 1.7 | 0:02:07 | 89.8 | 2.6 | 0:02:22 | 86.8 | 4.2 | 0:01:47 | 81.0 | 6.7 | 0:01:55 | 71.3 | 10.8 | 0:01:31 |
| | ✓ | ✗ | 85.2 | 1.7 | 0:02:07 | 50.4 | 2.6 | 0:02:22 | 21.3 | 4.2 | 0:01:47 | 14.1 | 6.7 | 0:01:55 | 10.8 | 10.8 | 0:01:31 |
| | ✗ | ✓ | 91.2 | 1.7 | 0:00:02 | 89.8 | 2.6 | 0:00:02 | 86.8 | 4.2 | 0:00:05 | 81.8 | 6.7 | 0:00:02 | 72.3 | 10.8 | 0:00:07 |
| | ✗ | ✗ | 50.3 | 1.7 | 0:00:02 | 16.4 | 2.6 | 0:00:02 | 10.5 | 4.2 | 0:00:05 | 11.7 | 6.7 | 0:00:02 | 10.2 | 10.8 | 0:00:07 |
| LayerActGrad | ✓ | ✓ | 90.6 | 2.8 | 0:02:01 | 87.8 | 6.1 | 0:02:04 | 85.0 | 10.6 | 0:01:37 | 80.6 | 15.5 | 0:01:43 | 71.2 | 21.3 | 0:02:02 |
| | ✓ | ✗ | 78.4 | 2.8 | 0:02:01 | 35.2 | 6.1 | 0:02:04 | 18.4 | 10.6 | 0:01:37 | 13.1 | 15.5 | 0:01:43 | 11.6 | 21.3 | 0:02:02 |
| | ✗ | ✓ | 90.5 | 2.9 | 0:00:06 | 88.4 | 5.7 | 0:00:05 | 85.3 | 10.1 | 0:00:10 | 81.6 | 15.1 | 0:00:10 | 72.2 | 20.2 | 0:00:05 |
| | ✗ | ✗ | 21.9 | 2.9 | 0:00:06 | 10.2 | 5.7 | 0:00:05 | 10.6 | 10.1 | 0:00:10 | 9.9 | 15.1 | 0:00:10 | 10.3 | 20.2 | 0:00:05 |
| LayerWeightNorm | ✓ | ✓ | 90.8 | 2.7 | 0:02:14 | 88.4 | 5.8 | 0:01:48 | 84.9 | 10.3 | 0:01:47 | 80.8 | 16.4 | 0:01:30 | 71.7 | 21.3 | 0:01:39 |
| | ✓ | ✗ | 81.8 | 2.7 | 0:02:14 | 46.7 | 5.8 | 0:01:48 | 20.4 | 10.3 | 0:01:47 | 12.0 | 16.4 | 0:01:30 | 11.0 | 21.3 | 0:01:39 |
| | ✗ | ✓ | 90.7 | 2.7 | 0:00:00 | 88.4 | 5.9 | 0:00:00 | 85.4 | 10.2 | 0:00:00 | 81.8 | 15.1 | 0:00:00 | 71.8 | 20.2 | 0:00:00 |
| | ✗ | ✗ | 25.4 | 2.7 | 0:00:00 | 9.9 | 5.9 | 0:00:00 | 9.4 | 10.2 | 0:00:00 | 9.9 | 15.1 | 0:00:00 | 9.8 | 20.2 | 0:00:00 |
| LayerSampling | ✓ | ✓ | 90.9 | 1.9 | 0:03:37 | 88.9 | 3.6 | 0:05:02 | 84.7 | 7.2 | 0:04:05 | 76.7 | 14.2 | 0:03:47 | 68.7 | 20.7 | 0:05:12 |
| | ✓ | ✗ | 79.2 | 1.9 | 0:03:37 | 32.7 | 3.6 | 0:05:02 | 14.0 | 7.2 | 0:04:05 | 11.8 | 14.2 | 0:03:47 | 10.3 | 20.7 | 0:05:12 |
| | ✗ | ✓ | 91.0 | 1.9 | 0:02:59 | 88.9 | 3.6 | 0:06:36 | 84.1 | 7.2 | 0:02:30 | 75.1 | 14.2 | 0:01:54 | 66.2 | 20.7 | 0:05:55 |
| | ✗ | ✗ | 26.0 | 1.9 | 0:02:59 | 11.9 | 3.6 | 0:06:36 | 11.0 | 7.2 | 0:02:30 | 9.5 | 14.2 | 0:01:54 | 9.5 | 20.7 | 0:05:55 |
| LayerGreedyFS | ✓ | ✓ | 90.6 | 2.6 | 0:38:44 | 88.2 | 5.3 | 8:43:34 | 84.6 | 9.9 | 0:10:17 | 80.2 | 15.1 | 0:06:57 | 71.4 | 17.4 | 0:03:16 |
| | ✓ | ✗ | 71.5 | 2.6 | 0:38:44 | 36.3 | 5.3 | 8:43:34 | 19.6 | 9.9 | 0:10:17 | 15.2 | 15.1 | 0:06:57 | 11.4 | 17.4 | 0:03:16 |
| | ✗ | ✓ | 90.8 | 2.6 | 0:30:11 | 88.6 | 5.7 | 0:36:54 | 85.5 | 9.7 | 0:19:09 | 80.7 | 16.4 | 0:04:44 | 73.0 | 21.3 | 0:02:05 |
| | ✗ | ✗ | 73.9 | 2.6 | 0:30:11 | 36.0 | 5.7 | 0:36:54 | 21.4 | 9.7 | 0:19:09 | 13.8 | 16.4 | 0:04:44 | 12.1 | 21.3 | 0:02:05 |
| LayerGreedyFS-fd | ✓ | ✓ | 90.6 | 2.6 | 6:54:48 | 88.4 | 5.6 | 0:21:51 | 85.0 | 9.9 | 4:19:50 | 80.0 | 15.2 | 0:07:27 | 70.6 | 17.4 | 0:06:58 |
| | ✓ | ✗ | 75.9 | 2.6 | 6:54:48 | 28.7 | 5.6 | 0:21:51 | 17.4 | 9.9 | 4:19:50 | 14.4 | 15.2 | 0:07:27 | 12.2 | 17.4 | 0:06:58 |
| | ✗ | ✓ | 90.7 | 2.6 | 1:09:55 | 88.6 | 5.7 | 0:25:45 | 85.6 | 9.3 | 0:18:38 | 81.1 | 15.3 | 0:11:40 | 74.3 | 22.0 | 0:04:07 |
| | ✗ | ✗ | 72.6 | 2.6 | 1:09:55 | 38.0 | 5.7 | 0:25:45 | 20.6 | 9.3 | 0:18:38 | 13.4 | 15.3 | 0:11:40 | 13.0 | 22.0 | 0:04:07 |

Table 5: Top-1 Accuracy % (Acc1), speedup ratio (SR), and pruning time (in hrs:mins:secs) of different pruning methods applied to VGG11 on CIFAR10, with different compression ratios $c$, with and without reweighting (rw) and fine-tuning (ft).

| Method | rw | ft | $c=2$ Acc1 | SR | time | $c=4$ Acc1 | SR | time | $c=8$ Acc1 | SR | time | $c=16$ Acc1 | SR | time | $c=32$ Acc1 | SR | time |
|---|---|---|---|---|---|---|---|---|---|---|---|---|---|---|---|---|---|
| AsymInChange | ✓ | ✓ | 89.7 | 1.9 | 31:38:06 | 89.7 | 2.1 | 30:35:13 | 89.6 | 2.5 | 23:54:22 | 89.5 | 5.0 | 22:38:37 | 81.4 | 8.3 | 19:01:51 |
| | ✓ | ✗ | 89.7 | 1.9 | 31:38:06 | 89.7 | 2.1 | 30:35:13 | 89.6 | 2.5 | 23:54:22 | 89.5 | 5.0 | 22:38:37 | 80.4 | 8.3 | 19:01:51 |
| | ✗ | ✓ | 90.1 | 1.9 | 34:15:16 | 90.1 | 2.1 | 33:33:05 | 89.3 | 2.5 | 24:58:50 | 89.1 | 4.7 | 18:59:56 | 87.6 | 6.8 | 16:52:36 |
| | ✗ | ✗ | 90.1 | 1.9 | 34:15:16 | 90.1 | 2.1 | 33:33:05 | 88.9 | 2.5 | 24:58:50 | 88.7 | 4.7 | 18:59:56 | 12.1 | 6.8 | 16:52:36 |
| SeqInChange | ✓ | ✓ | 90.1 | 1.9 | 30:29:41 | 90.1 | 2.1 | 30:31:14 | 90.0 | 2.5 | 23:25:17 | 89.8 | 5.0 | 25:04:38 | 81.9 | 8.3 | 23:54:55 |
| | ✓ | ✗ | 90.1 | 1.9 | 30:29:41 | 90.1 | 2.1 | 30:31:14 | 90.0 | 2.5 | 23:25:17 | 89.8 | 5.0 | 25:04:38 | 81.6 | 8.3 | 23:54:55 |
| | ✗ | ✓ | 90.1 | 1.9 | 32:17:12 | 90.1 | 2.1 | 31:13:02 | 89.3 | 2.5 | 23:51:44 | 89.3 | 4.7 | 22:31:17 | 87.7 | 6.8 | 20:23:53 |
| | ✗ | ✗ | 90.1 | 1.9 | 32:17:12 | 90.1 | 2.1 | 31:13:02 | 89.1 | 2.5 | 23:51:44 | 88.8 | 4.7 | 22:31:17 | 18.2 | 6.8 | 20:23:53 |
| LayerInChange | ✓ | ✓ | 90.1 | 1.9 | 27:27:33 | 90.1 | 2.1 | 29:43:24 | 90.0 | 2.5 | 20:45:59 | 89.8 | 5.0 | 29:00:23 | 85.3 | 8.3 | 18:56:08 |
| | ✓ | ✗ | 90.1 | 1.9 | 27:27:33 | 90.1 | 2.1 | 29:43:24 | 90.0 | 2.5 | 20:45:59 | 89.8 | 5.0 | 29:00:23 | 84.7 | 8.3 | 18:56:08 |
| | ✗ | ✓ | 90.1 | 1.9 | 36:41:49 | 90.1 | 2.1 | 34:49:36 | 89.2 | 2.5 | 29:05:17 | 89.1 | 4.7 | 21:39:32 | 87.8 | 6.8 | 20:45:04 |
| | ✗ | ✗ | 90.1 | 1.9 | 36:41:49 | 90.1 | 2.1 | 34:49:36 | 89.2 | 2.5 | 29:05:17 | 89.2 | 4.7 | 21:39:32 | 14.7 | 6.8 | 20:45:04 |
| ActGrad | ✓ | ✓ | 90.1 | 3.2 | 0:00:58 | 90.1 | 3.4 | 0:01:17 | 90.1 | 3.6 | 0:01:28 | 90.1 | 4.9 | 0:00:59 | 88.0 | 10.2 | 0:00:54 |
| | ✓ | ✗ | 90.1 | 3.2 | 0:00:58 | 90.1 | 3.4 | 0:01:17 | 90.1 | 3.6 | 0:01:28 | 90.1 | 4.9 | 0:00:59 | 83.2 | 10.2 | 0:00:54 |
| | ✗ | ✓ | 90.1 | 3.2 | 0:00:03 | 90.1 | 3.4 | 0:00:02 | 90.1 | 3.6 | 0:00:03 | 90.1 | 4.9 | 0:00:02 | 87.9 | 10.2 | 0:00:04 |
| | ✗ | ✗ | 90.1 | 3.2 | 0:00:03 | 90.1 | 3.4 | 0:00:02 | 90.1 | 3.6 | 0:00:03 | 90.1 | 4.9 | 0:00:02 | 56.8 | 10.2 | 0:00:04 |
| LayerActGrad | ✓ | ✓ | 89.7 | 2.0 | 0:03:53 | 90.0 | 2.1 | 0:02:17 | 89.7 | 2.5 | 0:01:03 | 89.5 | 4.6 | 0:00:48 | 87.7 | 8.4 | 0:01:03 |
| | ✓ | ✗ | 89.7 | 2.0 | 0:03:53 | 90.0 | 2.1 | 0:02:17 | 89.7 | 2.5 | 0:01:03 | 89.5 | 4.6 | 0:00:48 | 75.9 | 8.4 | 0:01:03 |
| | ✗ | ✓ | 90.1 | 1.9 | 0:00:05 | 90.1 | 2.0 | 0:00:06 | 90.1 | 2.6 | 0:00:05 | 89.5 | 5.0 | 0:00:06 | 87.2 | 5.5 | 0:00:08 |
| | ✗ | ✗ | 90.1 | 1.9 | 0:00:05 | 90.1 | 2.0 | 0:00:06 | 90.1 | 2.6 | 0:00:06 | 89.5 | 5.0 | 0:00:06 | 16.0 | 5.5 | 0:00:08 |
| LayerWeightNorm | ✓ | ✓ | 90.1 | 1.8 | 0:01:52 | 90.1 | 2.0 | 0:00:53 | 90.0 | 2.5 | 0:00:46 | 89.8 | 5.0 | 0:01:08 | 88.1 | 8.3 | 0:00:52 |
| | ✓ | ✗ | 90.1 | 1.8 | 0:01:52 | 90.1 | 2.0 | 0:00:53 | 90.0 | 2.5 | 0:00:46 | 89.8 | 5.0 | 0:01:08 | 80.9 | 8.3 | 0:00:52 |
| | ✗ | ✓ | 90.1 | 1.8 | 0:00:00 | 90.1 | 2.0 | 0:00:01 | 90.1 | 2.6 | 0:00:01 | 89.6 | 5.2 | 0:00:01 | 87.0 | 12.2 | 0:00:01 |
| | ✗ | ✗ | 90.1 | 1.8 | 0:00:00 | 90.1 | 2.0 | 0:00:01 | 90.1 | 2.6 | 0:00:01 | 89.6 | 5.2 | 0:00:01 | 11.6 | 12.2 | 0:00:01 |
| LayerSampling | ✓ | ✓ | 90.1 | 3.6 | 0:16:44 | 90.1 | 3.6 | 0:18:20 | 90.1 | 4.2 | 0:14:59 | 89.2 | 5.6 | 0:15:04 | 86.5 | 14.6 | 0:21:26 |
| | ✓ | ✗ | 90.1 | 3.6 | 0:16:44 | 90.1 | 3.6 | 0:18:20 | 90.1 | 4.2 | 0:14:59 | 89.2 | 5.6 | 0:15:04 | 75.0 | 14.6 | 0:21:26 |
| | ✗ | ✓ | 90.0 | 3.7 | 0:16:18 | 90.0 | 3.7 | 0:16:48 | 89.9 | 4.2 | 0:19:04 | 89.0 | 5.6 | 0:15:04 | 86.3 | 14.6 | 0:15:47 |
| | ✗ | ✗ | 90.0 | 3.7 | 0:16:18 | 90.0 | 3.7 | 0:16:48 | 89.9 | 4.2 | 0:19:04 | 50.5 | 5.6 | 0:15:04 | 13.7 | 14.6 | 0:15:47 |
| LayerGreedyFS | ✓ | ✓ | 89.3 | 2.0 | 23:30:16 | 83.3 | 2.9 | 7:50:23 | 74.2 | 4.8 | 5:12:07 | 40.3 | 10.9 | 6:01:53 | 31.2 | 18.3 | 1:51:43 |
| | ✓ | ✗ | 89.3 | 2.0 | 23:30:16 | 83.3 | 2.9 | 7:50:23 | 74.2 | 4.8 | 5:12:07 | 35.9 | 10.9 | 6:01:53 | 16.1 | 18.3 | 1:51:43 |
| | ✗ | ✓ | 89.2 | 2.0 | 9:49:18 | 88.7 | 2.2 | 6:56:45 | 87.2 | 2.8 | 4:10:47 | 84.4 | 4.8 | 2:40:57 | 46.4 | 7.4 | 2:36:44 |
| | ✗ | ✗ | 89.1 | 2.0 | 9:49:18 | 88.6 | 2.2 | 6:56:45 | 82.4 | 2.8 | 4:10:47 | 63.7 | 4.8 | 2:40:57 | 19.1 | 7.4 | 2:36:44 |
| LayerGreedyFS-fd | ✓ | ✓ | 89.4 | 2.0 | 12:06:03 | 86.8 | 2.7 | 10:22:33 | 69.1 | 4.0 | 6:00:33 | 44.3 | 13.0 | 3:27:58 | 31.1 | 13.0 | 3:35:15 |
| | ✓ | ✗ | 89.3 | 2.0 | 12:06:03 | 86.1 | 2.7 | 10:22:33 | 69.1 | 4.0 | 6:00:33 | 44.3 | 13.0 | 3:27:58 | 21.9 | 13.0 | 3:35:15 |
| | ✗ | ✓ | 89.4 | 2.0 | 12:42:58 | 88.6 | 2.6 | 7:40:45 | 87.1 | 3.3 | 5:04:15 | 85.5 | 4.9 | 3:45:22 | 63.5 | 11.5 | 1:59:09 |
| | ✗ | ✗ | 89.4 | 2.0 | 12:42:58 | 88.0 | 2.6 | 7:40:45 | 81.5 | 3.3 | 5:04:15 | 76.6 | 4.9 | 3:45:22 | 15.5 | 11.5 | 1:59:09 |