# OpenReview forum: "Data-Efficient Structured Pruning via Submodular Optimization"
_NeurIPS.cc/2022/Conference — NeurIPS 2022 Accept_

### Official Review · Reviewer_9DXV · 2022-06-21

**Rating:** 6
**Confidence:** 3
**Soundness:** 3 good
**Presentation:** 3 good
**Contribution:** 3 good

**Summary:**

Goal: effective and efficient structured pruning of a pre-trained NN-network if only small amount of unlabeled training data is available

Contributions:
1) The authors propose a new technique (called "principled data-efficient structured pruning") that alters the existing  "reweighting" method [Mariet and Sra, 2015]. Unlike [Mariet and Sra,2015]:
     * submodular optimization ... they  formulate the subset selection problem of  structured pruning as a weakly submodular maximization problem  and solve it approximately by greedy search
     * extended pruning: pruning of regular regions of neurons (e.g., channels) and three strategies for pruning of multliple layers
     * limited number of training data (cca. 1% of the original training data) with no labels and one-shot pruning (without fine-tuning)
2) Theoretical justification of the method and its performance (in the Supplementary material).
3) Experimental evaluation of a solid scope with promising results.


**Questions:**

1. In the abstract,  you write that "current structured pruning methods do not provide any performance guarantees". Later on, e.g., on lines 75-78,  you mention one such method. You should add something like "most of" into the abstract to be fair.

2. To confirm the novelty of your method, it would be nice to analyze in a more concrete way, what the  [Mariet and Sra,2015] method uses instead of submodular optimization and greedy search. Did you compare your approach to  [Mariet and Sra,2015]  exactly?

**Limitations:**

OK: The authors try to analyze both strengths and limitations of the paper in the experimental part.

**Strengths And Weaknesses:**

Strengths (significance and quality):
1) The problem of effective structured pruning of a pre-trained NN-network in the presented limited-data regime is important. Most of the existing structured pruning techniques require greater amount of training data and fine-tuning to work well. The proposed method (or even its particular parts) appears to be a valuable contribution with this respect.
2) The experimental evaluation of a solid scope offers promising (outstanding and stable) results.
3) Theoretical performance guarantee.
4) The authors try to be fair in their comparisons with concurrent techniques (e.g. the application of reweighting, various parameter settings...) and analyze also the weaknesses of their method.

Clarity:
1) The submission is relatively clearly written and easy to read (except the theoretical parts and the figures). If there was enough place, I would prefer to move more experimental results from the supplementary into the main paper.  The figures (e.g. Figure 1) are small and less comprehensible. It would maybe help to scale the graphs differently, to highlight the variants of the proposed method or to change the used colors.

Novelty:
1) The novelty of the contribution is slightly limited. The proposed method is based on previous work by [Mariet and Sra,2015]. The original method is altered using the principles of submodular optimization to be advantageous in the new context of  limited-data.
2) The related work is cited and addressed adequately.

---

> ### Author Response · Authors · 2022-08-02
> **Response to Reviewer 9DXV**
>
> Thank you for your positive review and the helpful suggestions for improving its clarity and presentation. We will increase the size of Figures in the camera-ready version. We address below your questions.
>
> 1. We modified the abstract as suggested.
>
> 2. As explained in our related work section (see lines 85-88), the method proposed in [Mariet and Sra, 2015] selects a subset of diverse neurons to keep in a given layer by sampling from a Determinantal Point Process (DPP), then applied reweighting at the end for the fixed choice of neurons, in the same fashion we apply reweighting to other pruning methods in our experiments. That is they are solving $\min_{\tilde{W}^{\ell+1} \in \mathbb{R}^{n_\ell \times n_{\ell+1}}} \\| A^\ell W^{\ell+1} - A^\ell\_S \tilde{W}^{\ell+1}  \\|_F^2$, with $S \sim \mathcal{P}$, where $\mathcal{P}$ is a DPP probability measure over the set of neurons in the layer. This is very different from our approach which selects neurons that minimize directly the reconstruction error after reweighting, i.e., we are solving $\min\_{|S| \leq k, \tilde{W}^{\ell+1} \in \mathbb{R}^{n\_\ell \times n\_{\ell+1}} } \\| A^\ell W^{\ell+1} - A^\ell\_S \tilde{W}^{\ell+1}  \\|_F^2$ (note the optimization over the set $S$). We did not compare with the method of [Mariet and Sra, 2015], as their experimental results show that the advantage of their method is mostly due to reweighting: In Figure 4 in [Mariet and Sra, 2015], their proposed method is only slightly better than random pruning + reweighting. In contrast, our experimental results (see Figure 1) show that the performance of our method is not only due to reweighting; our method outperforms other baselines (including random pruning) even when reweighting is applied to them too. Note also that [Mariet and Sra, 2015] use the full training data in their method, only consider pruning neurons, and only one layer at a time.

---

### Official Review · Reviewer_aJkM · 2022-07-09

**Rating:** 7
**Confidence:** 5
**Soundness:** 3 good
**Presentation:** 4 excellent
**Contribution:** 4 excellent

**Summary:**

The paper proposes a neural network node pruning method and shows that the objective is essentially a form of weakly-submodular function optimization. Therefore, the pruning of a single layer can be solved using the greedy algorithm with a theoretical guarantee. The paper also shows that using a limited number of data, the proposed method is able to achieve the best performance compared to baselines.

**Questions:**

1. Can authors estimate the weakly-submodular factors gamma in the experiments and report the corresponding guarantees? Can authors also discuss high tight are the estimates of such factors provided in Proposition 3/4?

2. Can authors provide wall clock running time of the optimization algorithm on the experiments and discuss the limitations of the complexity?

3. For multiple layer pruning, it seems that the changes of one layer could affect the pruning of other layers. Can the authors provide some insights about the empirical behaviors of convergence of the proposed methods?

**Limitations:**

Yes.

**Strengths And Weaknesses:**

Strengths:
1. The paper draws a very intriguing and solid connection between neural network pruning and submodular optimization. More specifically, the weakly-submodular optimization factor is closely related to the activation matrix.

2. The paper studies the problem in a comprehensive manner, including pruning of regular regions of neurons, strategies of pruning multiple layers, and speed-up tricks for the submodular optimization.

3. The proposed method empirically achieves the best performance compared to baselines on some network structures under a limited number of samples.

Weaknesses:
1. The proposed approach has relatively high computational complexity. It seems that scaling would be a problem for scenarios such as larger network structures or utilizing a large number of data samples for pruning.

2. Given the limitation of the complexity, I guess that the empirical experiments could be hardly extended to more complex datasets (e.g., ones with larger image dimensions) or larger network structures.

3. The proposed framework only works for one layer. For pruning of multiple layers the paper proposes some heuristics.

---

> ### Author Response · Authors · 2022-08-02
> **Response to Reviewer aJkM**
>
> Thank you for your positive review and the insightful questions. We address below your questions.
>
> 1. See our response about the empirical values of the submodularity ratio in our general comment.
> We added a discussion on the tightness of the lower bounds in Propositions 3.1 and 4.1 in Appendix F. We summarize here: There are cases where these bounds are tight. For example, if $A^\ell$ is the identity matrix, then both $F$ and $G$ are submodular, hence their corresponding $\gamma_{U, k}=1$ for all $U$ and $k$, and the lower bounds in both Proposition 3.1 and 4.1 are also equal to one. A more interesting example is when $A^\ell$ has columns with equal norm, and $W^{\ell+1}$ have all columns equal to the eigenvector of $(A^\ell_S)^\top A_S^\ell$ corresponding to the smallest eigenvalue over all sets $|S| \leq k$ (see details in Section D). On the other hand, there are also cases where these bounds are not tight. In particular, there are cases where the lower bound on $\alpha_{|U|+k-1}$ in Proposition C.4 (stronger notion of approximate submodularity) is larger than the lower bound on $\gamma_{U, k}$ in Proposition 3.1, which implies that the latter is not tight since $\gamma_{U, k} \geq \alpha_{|U|+k-1}$ (see Section C.1). For example, if all rows of $A^\ell W^{\ell+1}$ are linearly independent, but there exists $2k$ columns of $A^\ell$ which are linearly dependent, then the bound in Proposition 3.1 is zero while the one in  Proposition C.4 is not. These borderline cases are unlikely to occur in practice. Whether we can tighten these lower bounds based on realistic assumptions on the weights and activations is an interesting future research question.
>
> 2. See our response about the computational cost of our methods in our general comment.
>
> 3. Indeed pruning one layer will affect all subsequent layers. The sequential variants of our method takes this effect into account. Can you please clarify what you mean by the "empirical behaviors of convergence"?
> See our additional theoretical guarantees in our general comment where we provided a bound, with exponential convergence rate, on the difference between the original model output and the output after pruning one or multiple layers.
>
>  > The proposed framework only works for one layer. For pruning of multiple layers the paper proposes some heuristics.
>
>  See our additional theoretical guarantees on pruning multiple layers in our general comment.

---

### Official Review · Reviewer_455Y · 2022-07-13

**Rating:** 8
**Confidence:** 4
**Soundness:** 3 good
**Presentation:** 3 good
**Contribution:** 3 good

**Summary:**

This paper propose a network pruning approach via sub modular optimization. The proposed method uses a greedy fashion to layer-wisely select neuron that improves the performance most. This paper shows that the returned solution given by the greedy algorithm is able to well approximate the optimal solution (of a NP hard problem). To reduce the cost, a computation-saving approach is also proposed. Empirical results shows that the approach is able to achieve good performance when the available training data at pruning is small.

**Questions:**

See above.

**Limitations:**

Yes

**Strengths And Weaknesses:**

Overall I find this paper quite interesting, especially its performance when there is limited number of training data at pruning. The theoretical guarantee is also stronger and does not require stronger assumptions. However, I do have some concerns:

1. It seems the proposed method in general is very similar to Ye at al.. Both methods are greedy forward selection and the different seems subtle. Could you give more discussion in terms of methodology?

2. Following up Q1, it seems that an main improvement over Ye at al is to reduce the computation cost. However, [1] also propose a new technical to reduce the cost and I was wondering how does your method compares with [1]?

3. What would max_{|S|\le k} F(S) looks like? I understand that it is NP hard but it would be interesting to show how this quantities looks like.

4. Can we empirically verify the \gamma_{U,k} as this is an important quantity?

[1] Greedy Optimization Provably Wins the Lottery: Logarithmic Number of Winning Tickets is Enough

---

> ### Author Response · Authors · 2022-08-02
> **Response to Reviewer 455Y**
>
> Thank you for your positive review and the insightful questions. We address below your questions.
> 1. Our method is different from [Ye et al, 2020-a] both in terms of selection criteria and greedy algorithm.
> [Ye et al 2020-a] selects neurons/channels to keep in a given layer that minimize the loss of the pruned network. More precisely, they are solving $\min_{\alpha \geq 0, \sum_i \alpha_i = 1} \sum_{i=1}^n [(f^\ell_\alpha(x_i) - y_i)^2]$ (though in their code they use cross-entropy loss instead of square loss), where $(x_i, y_i)$ are data points, $f^\ell_\alpha(x)$ is the output of the model where neurons in layer $\ell$ corresponding to $\alpha_i=0$ are pruned and the weights of the next layer are scaled by $\alpha_i n_\ell$, i.e., $A^\ell W^{\ell+1}$ is replaced by $n_\ell A^\ell \mathrm{Diag}(\alpha) W^{\ell+1}$ where $\mathrm{Diag}(\alpha)$ is the diagonal matrix with $\alpha$ as its diagonal.
> This is only similar to an $\ell_1$-relaxation of the selection problem we solve in the special case of a two layer network with a single output, and instead of optimizing the weights of the next layer like we do, they optimize how much to scale them, i.e., in this case their selection problem reduces to $\min_{\alpha \geq 0, \sum_i \alpha_i = 1} \\| A^\ell w^{\ell+1} - n_\ell A^\ell \mathrm{Diag}(\alpha) w^{\ell+1}\\|_2^2$.
> They use a greedy algorithm with Frank-Wolfe like updates to approximate it (see Section 12.1 therein where they explain the relation between their greedy algorithm and Frank-Wolfe algorithm). The difference between their greedy algorithm and the one we use (Algorithm 1), aside from the objective function being different, is that they allow the selection of elements already selected.
> The advantage of our method over [Ye et al 2020-a] is not limited to the cheaper cost. Their theoretical guarantee only holds for two layer networks, and it is with respect to an $\ell_1$-relaxation of their selection problem, and not the original discrete problem like ours. Empirically, our method significantly outperforms the method of [Ye et al 2020-a] in all settings we consider.
>
> 2. Thank you for bringing the work of [Ye et al, 2020-b] to our attention. We discuss how our method compares to the two pruning methods, Greedy Global imitation and Greedy Local imitation, proposed therein. Greedy Global imitation is the same method from [Ye et al 2020-a] but with an additional approximation technique which reduces the cost of pruning one layer from $O(k n_\ell)$ to $O(k)$ forward passes through the full network. This is still more expensive than the cost of our method which is equivalent to $O(1/\epsilon)$ forward passes through only the layer being pruned, if using the fast Greedy algorithm from [Li et al., 2022] (see Section 3.3). Greedy Local imitation is closer to our approach, as it selects neurons/channels to keep in a given layer that minimize the change in the input to the next layer, but it also solves an $\ell_1$-relaxation of the selection problem and only optimize the scaling of the next layer weights instead of the weights directly, i.e., they solve $ \min_{\alpha \geq 0, \sum_i \alpha_i = 1} \\| A^\ell w^{\ell+1} - n_\ell A^\ell \mathrm{Diag}( \alpha ) w^{\ell+1} \\|_2^2 $. They again use a similar greedy algorithm with Frank-Wolfe like updates as in [Ye et al 2020-a]. Although the problem they solve is simpler than ours, the cost of pruning one layer using their method is still more expensive than ours: $O( k n_\ell n\_{\ell+1} n) $ vs $O((n\_\ell)^2 (n\_{\ell+1}+n+k)/\epsilon)$.
> They also provide bounds on the difference between the output of the original network and the pruned one, with exponential convergence rate for Greedy Local imitation, and  $O(1/k^2)$ rate for the Greedy Global imitation. We can derive similar guarantees with exponential convergence rate for our method; see our additional theoretical guarantees in our general comment for details.
> In terms of empirical comparison, the results in [Ye et al, 2020-b] show that their global method typically outperforms their local one. So although we did not directly compare with their local method empirically, we expect our method to do better, since it outperforms their global method.
>
>  [Ye et al, 2020-a] M. Ye, C. Gong, L. Nie, D. Zhou, A. Klivans, and Q. Liu. Good subnetworks provably exist: Pruning via greedy forward selection. ICML, 2020.
>
>  [Ye et al, 2020-b] M. Ye, L. Wu, and Q. Liu. Greedy optimization provably wins the lottery: Logarithmic number of winning tickets is enough. Advances in Neural Information Processing Systems, 33:16409–16420, 2020.
>
> We added this detailed comparison with [Ye et al, 2020-a,b] to Appendix A.

---

> > ### Author Response · Authors · 2022-08-02
> > **Response to Reviewer 455Y - Part 2**
> >
> > 3. The function $F$ (stated in Eq. 2) can also be written as  $F(S)  = \\| A^\ell W^{\ell+1}\\|_F^2 -  \\| (  A^\ell - \mathrm{proj}_S(A^{\ell})) W^{\ell+1}\\|_F^2, $ where $\mathrm{proj}_S(A^{\ell})$ is the matrix whose columns are the projections of the corresponding columns of $A^{\ell}$ onto the column space of $A^{\ell}\_S$. Hence, the optimal solution of $\max\_{|S| \leq k} F(S)$ minimizes the weighted error between the columns of $A^{\ell}$ and their projections. We hope this provides more insight about this problem. It is not possible to provide a closed form solution for the optimal $S$.
> >
> > 4. See our response about the empirical values of the submodularity ratio in our general comment.

---

> > > ### Comment · Reviewer_455Y · 2022-08-07
> > > **Thanks for addressing my questions.**
> > >
> > > I read the author response and find it properly addresses my concerns. I recommend the author to include some important part in to the next version. I raise my score by 1 and recommend an acceptance for this submission.

---

### Official Review · Reviewer_ULbp · 2022-07-14

**Rating:** 6
**Confidence:** 3
**Soundness:** 3 good
**Presentation:** 3 good
**Contribution:** 3 good

**Summary:**

The authors propose a layerwise pruning method that formulates the problem of eliminating neurons as a weakly submodular optimization problem for which the well-known greedy algorithm gives an approximation guarantee.  They illustrate the practical performance of three different variants of their strategy when extended to prune the entire network on a variety of tasks.

**Questions:**

It would be nice to see some discussion of the practical computational cost among the various methods in the main text.  While I don't see the cost of this method as overly limiting, it is nice to get a sense of how taxing the methods really are.

Also, some thoughts about why the approach seems to do much better in some regimes while only comparable in others?

**Limitations:**

Some limitations were discussed, though I don't see the focus on the data limited regime as a limitation of the methodology necessarily.  Or am I missing something important?

**Strengths And Weaknesses:**

Originality: The idea of formulating the pruning problem to take advantage of weak submodularity is novel to me.  Although it does build somewhat crucially on existing work.

Quality:  The technical and experimental results seem to be well-executed to the best of my assessment.  One can always add more competitors and try on a wider variety of architectures, but I found the selected experiments to be illustrative and compelling.

Clarity:  The exposition was clear overall though I found Figure 1 tough to read even after significant zooming.

Significance:  While it may be more expensive than some other approaches, the cost of this procedure is typically only born once.  So, I can see this as being a very useful tool in practice and may spur pruning research in novel directions.

---

> ### Author Response · Authors · 2022-08-02
> **Response to Reviewer ULbp**
>
> Thank you for your positive review and the insightful questions. We address below your questions.
>
> > It would be nice to see some discussion of the practical computational cost among the various methods in the main text.
>
> See our response about the computational cost of our methods in our general comment.
>
> >  some thoughts about why the approach seems to do much better in some regimes while only comparable in others?
>
> Our approach performs better in settings where $\gamma_{\hat{S}, k}$ is larger. This could explain for example why it performs relatively the best on LeNet model. Though this not the only factor affecting performance; using random patches for example increases the range of $k$ where $\gamma_{\hat{S}, k}$ is non-zero, but deterioates the accuracy achieved by our method. See the discussion about the empirical values of the submodularity ratio in our general comment.
>
> > I found Figure 1 tough to read even after significant zooming.
>
> We will increase the size of Figures in the camera-ready version.

---

> > ### Comment · Reviewer_ULbp · 2022-08-08
> > **Thanks for the additional details**
> >
> > I agree that the focus of the paper isn't on speed, but it's always nice to get a sense of the time scales involved.

---

### Author Response · Authors · 2022-08-02
**General comment to all reviewers**

We thank all reviewers for their careful reading of our paper and their helpful suggestions.
We have uploaded a revision of our paper based on the reviewers feedback.
We address here questions and concerns shared by several reviewers:


1. **Additional theoretical guarantees (Reviewers aJkM & 455Y):**
We thank Reviewer **455Y** for pointing the work of [Ye et al, 2020-b] to us. To better answer the question of how our method compares to this work, we extended our theoretical guarantees to obtain ones of similar type as the guarantees in [Ye et al, 2020-b]. We included them in Appendix D. We will move them (without the proofs) to the main text in the camera-ready version. We state the results here: we show that the change in input to the next layer induced by pruning with our method decays with exponentially fast rate:
$$\\| A^\ell W^{\ell+1} - A^\ell_{\hat{S}} \hat{W}^{\ell+1}  \\|_F^2 \leq e^{- \gamma\_{\hat{S}, n_\ell} {k}/{n_\ell}} \\| A^\ell W^{\ell+1}\\|_F^2, $$
where $\hat{S}$ is the output of the Greedy algorithm, and $\hat{W}^{\ell+1}$ are the corresponding optimal weights (see Eq. (4)).
Moreover, if we assume as in [Ye et al, 2020-b] that the function $H$ corresponding to all layers coming after layer $\ell$ is Lipschitz continuous, we can show that the difference between the original model output $y$ and the output $y^{\hat{S}}$ after layer $\ell$ is pruned using our method, also decays with exponentially fast rate:
$$\sum\_{i=1}^n (y\_i - y^{ \hat{S} }\_i)^2  \leq e^{- \gamma\_{\hat{S}, n_\ell} k/n_\ell} \\| H \\|\_{\text{Lip}}^2  \\| A^\ell W^{\ell+1} \\|_F^2, $$
which matches the convergence rate achieved by the local imitation method in [Ye et al, 2020-b] (see Theorem 1 therein).
Under the same assumption, we show that the difference between the original model output $y$ and the output $y^{\hat{S}\_L}$ after pruning $L$ layers with the sequential variant of our method (SeqInChange) also admits an exponential convergence rate:
$$ \sum\_{i=1}^n (y\_i - y^{ \hat{S}\_L }\_i)^2 \leq \sum\_{\ell=1}^L e^{- \gamma\_{ \hat{S}\_\ell, n\_\ell } {k\_\ell}/{n\_\ell} } \\| H_\ell \\|\_{\text{Lip}}^2  \\| B^\ell W^{\ell+1} \\|_F^2,$$
where recall that $B^\ell$ are the updated activations of layer $\ell$ after previous layers $1$ to $\ell-1$ are pruned. This again matches the convergence rate achieved by the local imitation method in [Ye et al, 2020-b] (see Theorem 6 therein).

We can obtain an even stronger bound for the asymmetric sequential variant of our method (AsymInChange):
$$\sum_{i=1}^n (y_i - y^{\hat{S}\_L}\_i)^2 \leq \sum_{\ell=1}^L \prod_{\ell'=\ell+1}^L (1- e^{- \gamma_{\hat{S}\_{\ell'}, n_{\ell'}} {k_{\ell'}}/{n_{\ell'}}}) e^{- \gamma\_{\hat{S}\_\ell, n_\ell} {k_\ell}/{n_\ell}} \\| H_\ell\\|_{\text{Lip}}^2  \\| A^\ell W^{\ell+1}\\|_F^2.$$
The last two results address Reviewer **aJkM** comment on our guarantees only applying to pruning one layer.

 [Ye et al, 2020-b] M. Ye, L. Wu, and Q. Liu. Greedy optimization provably wins the lottery: Logarithmic number of winning tickets is enough. Advances in Neural Information Processing Systems, 33:16409–16420, 2020.

---

> ### Author Response · Authors · 2022-08-02
> **General comment to all reviewers - Part 2**
>
> 2. **Empirical values of the submodularity ratio $\gamma\_{U, k}$ (Reviewers aJkM & 455Y):**
> Computing the lower bounds on the submodularity ratio $\gamma\_{U, k}$ given in Proposition 3.1 and 4.1 is NP-Hard, as stated in [Das and Kempe, 2011] ($\min\_{\\| z \\|_2 = 1, \\| z \\|_0 \leq |U|+k} \\|A^\ell z\\|_2^2$ corresponds to $\lambda\_{\min}(C, |U| +k)$ in their notation). One simple lower bound that can be obtained from the eigenvalue interlacing theorem is $\gamma\_{U, k} \geq \frac{\lambda\_{\min}((A^\ell)^\top A^\ell)}{\lambda\_{\max}((A^\ell)^\top A^\ell)}.$ However, such bound is too loose as it is often equal to zero. For this reason, we focused in the paper on when our lower bounds on $\gamma\_{\hat{S}, k}$ are non-zero (see discussions on lines 143-146 and lines 205-213): the lower bounds in Propositions 3.1 and 4.1 are non-zero if any $2k$ and $2 k r_h r_w$ columns of $A^\ell$ are linearly independent, respectively.  We report below an upper bound on $k$ for which these conditions hold, based on the rank of the activation matrix $A^\ell$, in each pruned layer in the three models we used in our experiments.
>
> | Model    | upper bound on $k / n_\ell$  (all patches, $n=512$) |
> |----------|---------------------------------------------------------------------------------|
> | LeNet    | conv1: 0.37, conv2: 0.50, fc1: 0.46, fc2: 0.49  |
> | VGG11    | features.0: 0.22, features.4: 0.39, features.8: 0.25,  features.11: 0.28, features.15: 0.06, features.18: 0.03,  features.22: 0.01, classifier.0: 0.5, classifier.3: 0.5 |
> | ResNet56 | layer1.0.conv1: 0.12, layer1.1.conv1: 0.15, layer1.2.conv1: 0.22,  layer1.3.conv1: 0.22, layer1.4.conv1: 0.30, layer1.5.conv1: 0.28,  layer1.6.conv1: 0.44, layer1.7.conv1: 0.35, layer1.8.conv1: 0.15,  layer2.0.conv1: 0.48, layer2.1.conv1: 0.47, layer2.2.conv1: 0.48,  layer2.3.conv1: 0.47, layer2.4.conv1: 0.48, layer2.5.conv1: 0.47,  layer2.6.conv1: 0.48, layer2.7.conv1: 0.47, layer2.8.conv1: 0.48,  layer3.0.conv1: 0.47, layer3.1.conv1: 0.49, layer3.2.conv1: 0.46,  layer3.3.conv1: 0.48, layer3.4.conv1: 0.48, layer3.5.conv1: 0.49,  layer3.6.conv1: 0.48, layer3.7.conv1: 0.48, layer3.8.conv1: 0.50 |
>
> As explained (on lines 205-213), in some layers the linear independence condition required for convolution layers (Proposition 4.1) only holds for very small $k$, due to the correlation between patches which overlap. This can be avoided in most layers by sampling $r_h r_w$ random patches from each image (to ensure $A^\ell$ is a tall matrix), instead of using all patches. We report below the corresponding upper bounds on $k$ in this setting for the VGG11 model.
>
> | Model    | upper bound on $k / n_\ell$  (random patches, $n=512$) |
> |----------|---------------------------------------------------------------------------------|
> | VGG11    | features.0: 0.38, features.4: 0.43, features.8: 0.29, features.11: 0.31, features.15: 0.15, features.18: 0.08, features.22: 0.1, classifier.0: 0.5, classifier.3: 0.5 |
>
>
> The upper bounds are indeed larger than the ones obtained with all patches for most layers. However, some layers (e.g., features.15, 18, 22) have a very small feature map size (respectively $4\times 4, 4\times 4, 2\times 2$) so that even the small number of random patches will have significant overlap. Our experiments with random patches on VGG11 yielded worst results, so we chose to use all patches.
> Note that our lower bounds on $\gamma\_{\hat{S}, k}$ are not necessarily tight (see our response to Reviewer **aJkM**'s first question for more details). Hence, if $k$ is outside these ranges, our lower bound on $\gamma\_{\hat{S}, k}$ is zero, but not necessarily $\gamma\_{\hat{S}, k}$ itself; indeed in our experiments our methods still perform well in these cases. We added this more detailed discussion in Appendix E.

---

> > ### Author Response · Authors · 2022-08-02
> > **General comment to all reviewers - Part 3**
> >
> > 3. **Computational cost of our methods (Reviewers ULbp & aJkM):**
> >  We report below the pruning time (in hrs:mins:secs) of the three variants of our methods for different fraction values on the experiments of Section 6 (Figure 1).
> >
> > | Model | Method          | 10%     | 25%     | 50%             |
> > |:------|:----------------|:---------|:---------|:--------------|
> > | LeNet | AsymInChange  | 00:00:01 | 00:00:03 | 00:00:04 |
> > | LeNet | SeqInChange   | 00:00:01 | 00:00:02 | 00:00:05 |
> > | LeNet | LayerInChange | 00:00:01 | 00:00:02 | 00:00:04 |
> > | VGG11 | AsymInChange  | 10:28:09 | 18:06:55 | 40:08:32 |
> > | VGG11 | SeqInChange   | 17:23:44 | 23:47:09 | 36:33:56 |
> > | VGG11 | LayerInChange | 17:07:59 | 21:43:08 | 29:46:10 |
> > |ResNet56| AsymInChange  | 01:08:40 | 02:38:18 | 04:36:53 |
> > |ResNet56| SeqInChange   | 01:23:20 | 03:04:16 | 04:09:00 |
> > |ResNet56| LayerInChange | 01:20:23 | 03:06:15 | 05:29:59 |
> >
> > Our goal in these experiments was to show that our method outperforms other baselines in terms of accuracy vs compression rate. As such we did not focus on optimizing the computation time efficiency of our implementation. For example, our current implementation uses the classical Greedy algorithm. This can be significantly sped-up by switching to the faster Greedy algorithm from [Li et al., 2022].
> > In terms of theoretical computational complexity, as discussed in Section 3.3 (lines 173-181), the pruning time of our method per layer is equivalent to $O(k)$ forward passes through the layer being pruned, if using the classical Greedy algorithm (Algorithm 1), and $O(1/\epsilon)$ passes if using the faster Greedy algorithm from [Li et al., 2022]. However, in practice deep learning libraries like Pytorch provide highly optimized implementations of the convolution and linear layers. Obtaining similar speedups for our method would require significant engineering effort.

---

### Meta-Review · Area_Chair_FjAg · 2022-08-21

**Recommendation:** Accept
**Confidence:** Certain

**Metareview:**

The paper proposes a data-efficient structured pruning method, that for a given layer finds neurons/channels to prune with corresponding new weights for the next layer, that minimize the change in the next layer's input induced by pruning. This selection problem is formulated as a weakly submodular maximization problem, thus it can be provably approximated using the greedy algorithm. The proposed solution is interesting and practical as it requires limited-number of training data and no labels. The reviewers found the authors' response convincing, however the authors are strongly encouraged to incorporate the clarifications provided in the rebuttal into the final version.

**Award:**

No

---

### Decision · Program_Chairs · 2022-09-14

Accept